Article 

# Nerve growth factor receptor limits inflammation to promote remodeling and repair of osteoarthritic joints

Lan Zhao [1,2], Yumei Lai [1], Hongli Jiao[1] & Jian Huang [1,2] ✉

Osteoarthritis (OA) is a painful, incurable disease affecting over 500 million people. Recent clinical trials of the nerve growth factor (NGF) inhibitors in OA patients have suggested adverse effects of NGF inhibition on joint structure. Here we report that nerve growth factor receptor (NGFR) is upregulated in skeletal cells during OA and plays an essential role in the remodeling and repair of osteoarthritic joints. Specifically, NGFR is expressed in osteochondral cells but not in skeletal progenitor cells and induced by TNFα to attenuate NF-κB activation, maintaining proper BMP-SMAD1 signaling and suppressing RANKL expression in mice. NGFR deficiency hyper-activates NF-κB in murine osteoarthritic joints, which impairs bone formation and enhances bone resorption as exemplified by a reduction in subchondral bone and osteophytes. In human OA cartilage, NGFR is also negatively associated with NF-κB activation. Together, this study suggests a role of NGFR in limiting inflammation for repair of diseased skeletal tissues.

Osteoarthritis (OA) is a painful, incurable disease affecting 500 million adults around the world[1]. Pathological changes of OA include multiple manifestations such as articular cartilage destruction, joint space narrowing, synovial hyperplasia, osteophyte formation, and subchondral bone sclerosis[2,3]. The hallmark of OA is articular cartilage degradation, which can be induced by aging, injury, overloading, or abnormal inflammation[4–6]. The bony changes, such as sclerosis in subchondral bone and osteophytes at the periphery or margin of joints, are also prominent radiographic features of OA, indicative of the later stages of the disease[2,7]. Both subchondral sclerosis and osteophytes are thought to be a spontaneous response of the diseased joint to mechanical overload due to the loss of cartilage[2,8]. In addition, modification of these bony remodeling processes could potentially treat OA, as antiresorptive agents offer beneficial effects on articular cartilage protection[2].

Pain is the most prominent complaint from OA patients and an overwhelmingly influential factor in causing disability[9]. Nerve growth factor (NGF) was the first growth factor discovered to be essential for the growth and survival of neurons[10]. A member of the neurotrophin family, NGF has been found to be significantly upregulated in the joints of OA patients[11], which is believed to be required for the sensitization of nociceptors, the sensory neurons and their endings that detect signals from peripheral tissues. Clinical trials have shown that monoclonal antibodies against NGF, such as tanezumab, offer significantly greater pain relief and improved physical function in OA patients but are associated with more prevalent joint safety events than the non-steroidal anti-inflammatory drug (NSAID) group[12,13]. The majority of joint safety events were described as rapidly progressive osteoarthritis (RPOA), which shows accelerated loss of joint space width or abnormal bone loss, destruction, or collapse[12,13]. Therefore, the safety risk of tanezumab may outweigh its benefit in relieving pain. For further improvement of new analgesics represented by NGF inhibitors, it would be important to gain a clear understanding of the mechanism underlying RPOA associated with NGF inhibition, which has yet to be identified. It cannot be explained by tanezumab-induced pain relief, as analgesia was similar in patients with and without RPOA[12]. Subchondral insufficiency fracture and atrophic OA, a subset of OA showing no or tiny osteophytes, may increase the risk of RPOA[14,15], suggesting that RPOA is associated with weakened bone formation.

[1]Department of Orthopedic Surgery, Rush University Medical Center, Chicago, IL, USA. [2]These authors contributed equally: Lan Zhao, Jian Huang.
✉e-mail: jian_huang@rush.edu

To activate the NGF signaling pathway, NGF binds to its transmembrane receptors, TrkA (tropomyosin receptor kinase A), the high-affinity catalytic receptor for NGF, or p75NTR, also known as nerve growth factor receptor (NGFR) or CD271, the one with a lower NGF-binding capacity[16]. The binding of NGF to its receptors leads to the autophosphorylation of the receptors, initiating the downstream signaling cascades such as MAPK and PI3K-AKT, which are essential to neuronal survival and growth, thereby sensitizing neurons and stimulating the growth of axons and dendrites[17–19]. Our previous results have demonstrated that NGF expression is significantly upregulated in periarticular tissues, especially in synovium and newly formed ectopic cartilage/bone, which may elicit neurite growth in the arthritic joint[20,21]. Large-scale RNA-Seq studies revealed that TrkA, the main NGF receptor in sensory neurons, is restricted to peripheral nociceptors, suggesting that its role may be specifically limited to nociceptor innervation[22].

In contrast, the expression profile of NGFR is more diverse and not restricted to neurons. Particularly, its expression in myeloid cells could be stimulated under pathological or inflammatory conditions[22]. In regards to the NGFR expression in skeletal cells, previous reports suggest that NGFR is expressed in a subset of mesenchymal cells or bone marrow stromal cells that show greater capacities of self-renewal, ex vivo hematopoiesis support and multipotent differentiation into osteoblasts, chondrocytes, or adipocytes[23–26]. Interestingly, the generally viable and fertile phenotype of NGFR null mice[27,28], which display abnormal reflexes or reduced innervation, seems to obscure whether NGFR has an indispensable function in skeletal tissues.

In this study, we identified nerve growth factor receptor (NGFR) as a prominent receptor that responds to NGF, a major growth factor associated with pain, in osteochondral joint cells. Our results established that the NGFR signaling initiates anti-inflammatory and anabolic reactions to repair and stabilize osteoarthritic joints, an important yet unappreciated role of NGFR in regulating non-neuronal skeletal cells and tissues during OA progression.

## Results

### NGF induction is associated with bony remodeling and inflammation during OA pathogenesis and progression

To delineate the pathological course of OA, we generated a mouse model of posttraumatic OA through surgical induction of destabilization of the medial meniscus (DMM)[29]. Two weeks after the DMM surgery, the joint displayed severe inflammation, as represented by synovial hyperplasia and monocyte infiltration (Fig. 1a), suggesting an acute response to traumatic surgery. At the time point of 4 weeks post-surgery, synovial inflammation still persisted, albeit with reduced severity, and bony changes, including subchondral sclerosis and synovial calcification, became evident (Fig. 1b, c), which underscores synovial inflammation and bony remodeling in OA progression[30].

In addition, we observed significant alterations related to neurite growth in osteoarthritic joints, as demonstrated by the increased levels of NGF (Fig. 1d) and βIII-tubulin, a specific neuronal marker in neurogenesis (Fig. 1e), particularly in hypertrophied synovium, similar to our previous reports[20,31]. Largely overlapping with NGF expression in synovium and newly formed ectopic cartilage/bone, inductions of RUNX2 and SMAD1, both pivotal transcription factors in osteogenesis[32], were remarkable in these osteochondral tissues (Fig. 1f–h). Thus, our results suggested that NGF induction is concurrent with multiple events, including inflammation and bony remodeling during the course of OA progression.

### Expressions of NGF receptors in skeletal cells

To clarify whether NGF induction has a direct role in skeletal pathophysiology, firstly we aimed to confirm whether there are significant expressions of the NGF receptors, TrkA and NGFR, in the skeletal cells. Among the 11 cell lines we examined, the 5 cell lines thought to be mesenchymal stem cells or early skeletal progenitor cells, including

C3H10T1/2 (mouse embryo fibroblast cell line)[33], ST2 (mouse bone marrow stromal cell line)[34], ATDC5 (mouse prechondrogenic stem cell line)[35], mouse CD45⁻ bone marrow stromal cells (BMSCs)[20], and human mesenchymal stem cells (hMSCs), did not express a detectable level of NGFR protein (Fig. 2a, b). Interestingly, mouse E14.5 limb bud cells, mouse perinatal articular chondrocytes (AC), rat chondrosarcoma (RCS) cells, human osteoblastic cell lines hFOB1.19 and Saos-2, and mouse progenitor cell line C2C12, a pluripotent cell line capable of differentiating into osteoblasts, have readily detectable levels of NGFR protein (Fig. 2a, b). CRISPR-mediated knockout or transcriptional activation (CRISPRa)[36] of Ngfr also validated the presence of both NGFR mRNAs and proteins in the cells (Fig. 2c, Supplementary Fig. 1a, b). We also compared NGFR proteins in dorsal root ganglia (DRG), hMSCs, and osteoblasts and found that NGFR protein is abundant in DRG (Supplementary Fig. 2). In contrast, the higher affinity NGF receptor, TrkA, also known as NTRK1, had undetectable protein levels in all skeletal cells tested (Fig. 2d). In addition, pan Trk proteins, including TrkA, TrkB and TrkC, were not detectable in all of the skeletal cells. Thus, our results suggest that TrkA plays a specialized role in neurons, whereas NGFR has more diverse functions in both neurons and skeletal cells.

We also examined the mRNA expression of NGF receptors. Generally, the mouse skeletal cells with a committed osteochondral fate, such as mouse ACs, E14.5 limb bud cells, and C2C12 cells, had higher expression of Ngfr compared to the early-stage cells, including BMSCs and C3H10T1/2, which had a negligible expression of Ngfr mRNAs (Fig. 2e). In the RCS cells, TrkA expression was also minimal and Ngfr was about 50-fold higher (Supplementary Fig. 1c). NGFR was barely detectable in human MSCs, but much more conspicuous in hFOB1.19 and Saos-2. The comparison between NGFR and TrkA showed that NGFR was much more abundant than TrkA (note their ratios to β-actin) in osteoblastic cells (Fig. 2f, g). Further, we examined NGFR protein levels in cultured articular chondrocytes and E14.5 limb bud cells with different passage numbers. In general, repeated passages will lead to dedifferentiation of the cells, which lose their chondrocytic or osteoblastic phenotype[37]. Our data demonstrated that NGFR protein levels were significantly reduced in dedifferentiated chondrocytes or limb bud cells (Fig. 2h, i), further confirming that NGFR is more predominantly expressed in the cells with committed osteochondral fates.

In addition, we utilized a published single-cell RNA-Seq (scRNA-Seq) dataset of human embryonic skeletogenesis[38] for analyses of gene expressions of NGFR and several skeletal cell markers. Clustering and visualization of the scRNA-Seq data demonstrated that cells from the human limb bud at Carnegie stage (CS) 13 (about 5 weeks post-conception, WPC) and human long bone at CS22 (about 8 WPC) can be grouped into 11 clusters in an integrative analysis (Fig. 3a). Notably, both CS13 and CS22 samples exhibited low frequencies (<5%) of non-skeletal cells (Fig. 3b), including EPCAM⁺ epithelial cells (cluster 8) in limb buds, SOX10⁺ Schwann cells (cluster 11) in long bone, and some shared cell types in both samples such as MYOG⁺ SIX1⁺ myoprogenitors/myocytes (clusters 5 and 9), GYPA⁺ erythrocytes (cluster 6), CDHS⁺ endothelial cells (cluster 7), and CD68⁺ PTPRC⁺ macrophages (cluster 10)[38]. In contrast, most of the cells in both samples were identified as skeletal cells (Clusters 0–4) due to their high expression levels of PRRX1 and PDGFRA, two skeletal progenitor cell markers[38,39]. In addition, these skeletal cells appeared to have extensive communications between different populations (Fig. 3c). The collective expression of NGFR in these skeletal cells was significantly upregulated in the sample of CS22 compared to that in CS13, while PRRX1 expression was similar between the two samples (Fig. 3d). Indeed, Cluster 2 that represents chondroblasts and chondrocytes expressing SOX9 and ACAN was exclusively detected in CS22 (Fig. 3b, e), suggesting that chondrogenesis did not occur in CS13, which expressed a minimal level of NGFR. Further, in each cluster of those skeletal progenitor populations (Clusters 0, 1, 3, 4) shared by the two samples, NGFR induction in CS22

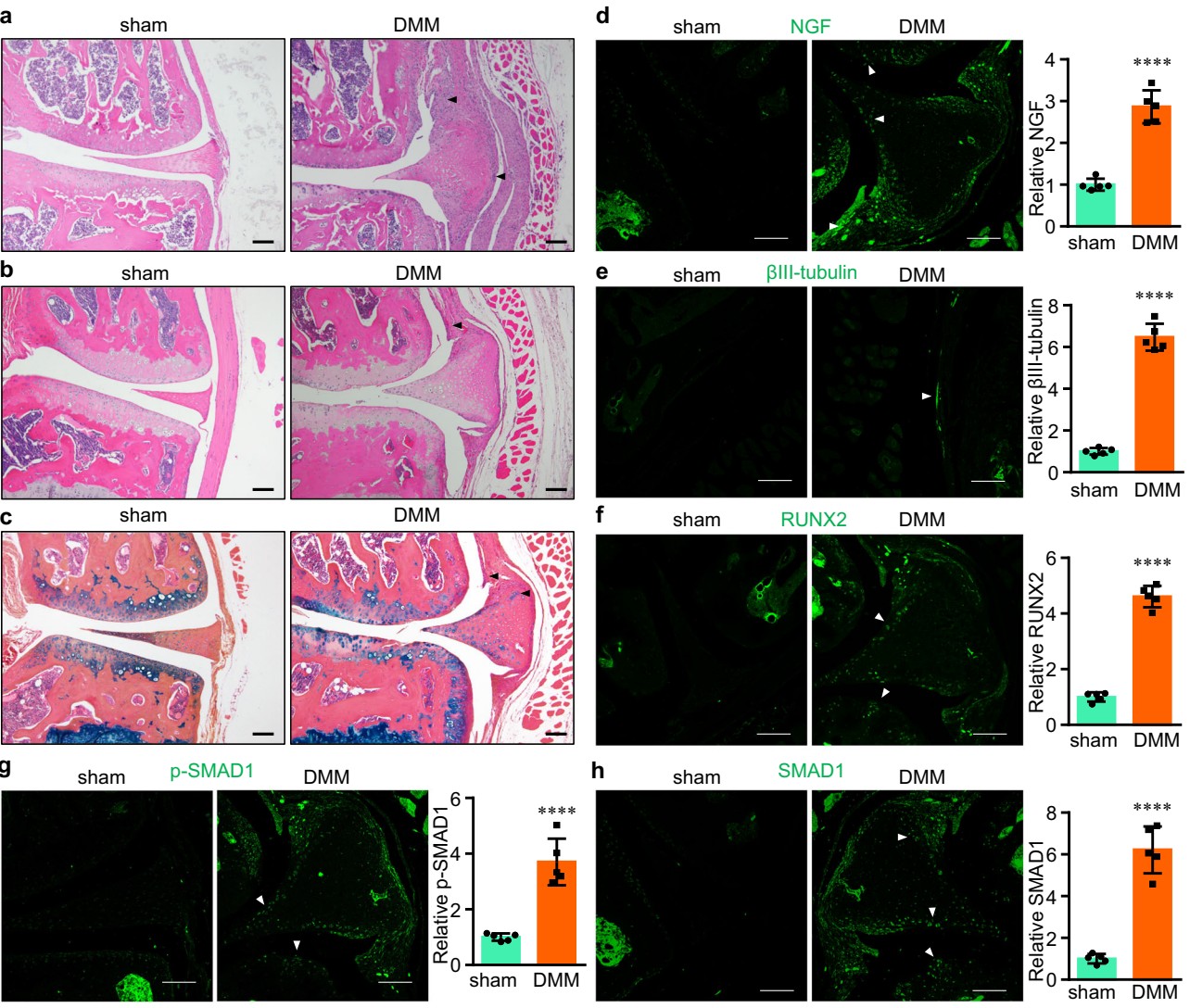

**Fig. 1 | Inflammation, neurogenesis, and bony remodeling during OA pathogenesis and progression. a, b** Representative histology images of osteoarthritic knee joint sections stained by hematoxylin and eosin. The joints were collected 2 (**a**) or 4 weeks (**b**) after the DMM surgery. Arrowheads, inflamed synovial tissues. **c** Representative histology images of osteoarthritic knee joint sections stained by alcian blue/hematoxylin & orange G. The joints were collected 4 weeks after the DMM surgery. Arrowheads, synovial hyperplasia and osteochondral formation. Scale bar, 100 µm. $n = 6$ joints. **d–h** Representative immunohistochemistry (IHC) images of osteoarthritic knee joint sections stained by individual antibodies as shown. The joints were collected 4 weeks after the DMM surgery. Arrowheads positively IHC signals. Scale bar, 100 µm. Data are presented as mean values ± SD. $n = 5$ independent experiments. Unpaired two-sided Student's $t$-test. ****$p < 0.0001$. Source data are provided as a Source Data file.

was consistent (Fig. 3e). Thus, our analysis suggested that *NGFR* is upregulated during the osteochondral differentiation of skeletal progenitor cells.

To confirm the in vivo expression of NGFR, we performed immunohistochemistry (IHC) to analyze the protein levels of NGFR in osteoarthritic joints. Interestingly, the NGFR protein level increased in osteoarthritic joints compared to that in naïve joints, especially in the newly formed ectopic osteochondrophytes (Supplementary Fig. 3a, b). Together, our results suggest that TrkA plays a specialized role in neurons, while NGFR has a non-neuronal role in cells with osteochondral commitment that responds to OA pathogenesis.

### Exploring the role of osteochondral NGFR

It has been established that skeletal cells expressing aggrecan give rise to not only chondrocytes but also the majority of osteoblasts and osteocytes during endochondral bone formation[40,41]. In addition, our scRNA-Seq analysis demonstrated that *Aggrecan* has significant expressions in skeletal progenitor cells (Fig. 3e). Therefore, *Acan-CreER*

could be useful for comprehensive analysis of osteochondral formation in the joint, a complex organ comprising multiple types of skeletal cells. Thus, we generated *Ngfr^Acan-CreER* (hereafter KO) mice, in which NGFR deficiency was induced through tamoxifen injection for five consecutive days when the mice were 15 days old. After five months, we collected the knee joints from both the control and *Ngfr^Acan-CreER* mice for radiographic and histologic analyses. Surprisingly, we did not find overtly appreciable differences in joint architectures comprising articular cartilage, subchondral bone, synovium, and meniscus between the two groups (Supplementary Fig. 4a–c), suggesting that tamoxifen pulsed at the age of 2 weeks did not fundamentally alter joint structure and homeostasis 5 months later, or NGFR may not be essential for normal joint development and homeostasis.

To further investigate whether NGFR is required by non-neuronal joint cells during OA, we injected the 15-day-old mice with tamoxifen for 5 consecutive days and performed DMM to induce OA pathogenesis when they reached 3 months of age. As expected, the DMM surgery profoundly altered joint architecture in both the control and KO

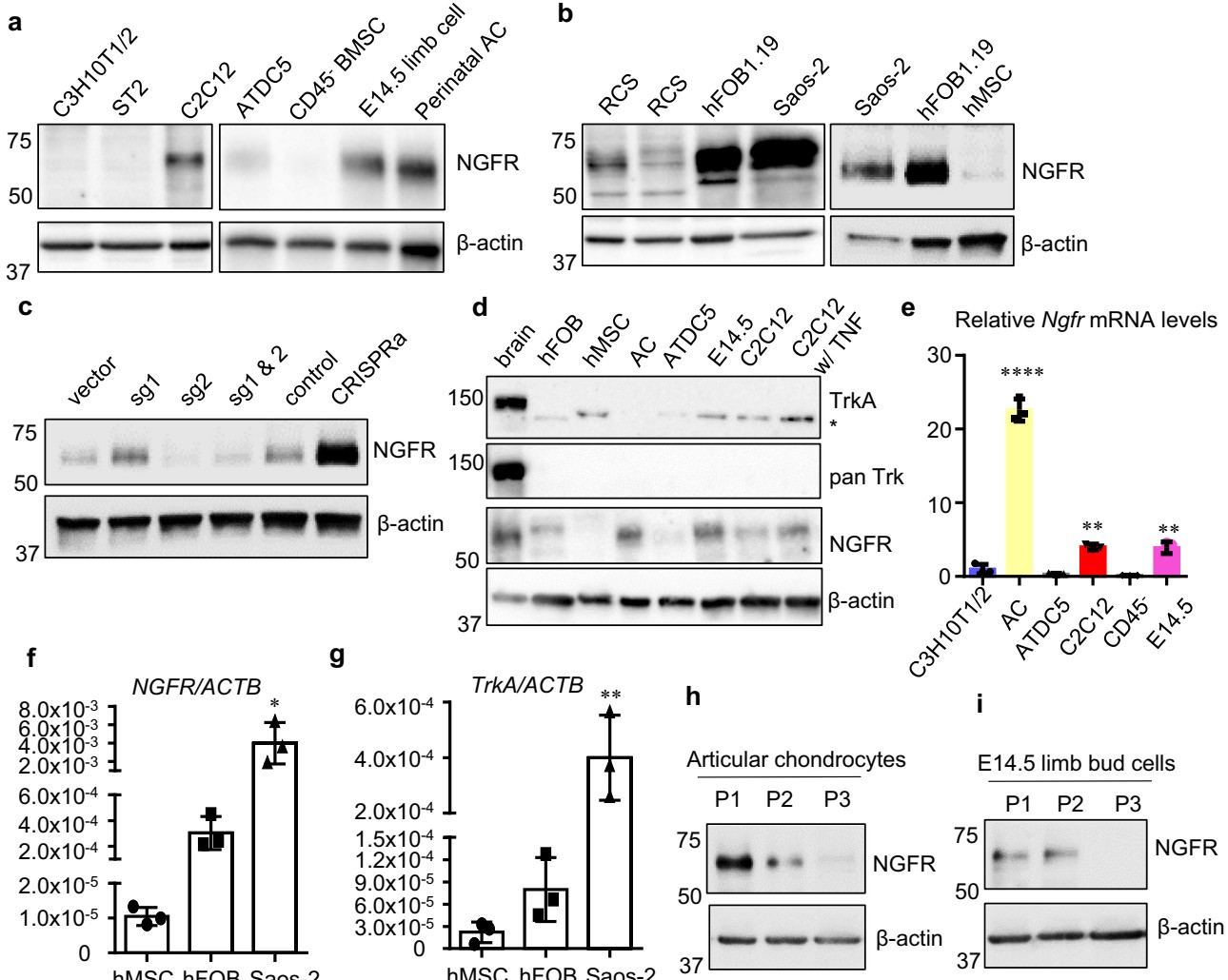

**Fig. 2 | NGFR is significantly upregulated in skeletal cells with committed osteochondral fates compared to progenitor cells at earlier stages. a, b** Western results of NGFR in mouse cells (**a**) and in multiple human cell lines and RCS cells (**b**). $n = 3$ independent experiments. **c** Perturbations of NGFR validated the anti-NGFR antibody and confirmed NGFR proteins in C2C12. sg single guide RNA for CRISPR to mutate *Ngfr*. CRISPRa indicates CRISPR-mediated transcriptional activation of *Ngfr*. Control is the empty vector for CRISPRa. Note that sg1 targeting the promoter increased NGFR expression. $n = 3$ independent experiments. **d** TrkA had no detectable protein levels in skeletal cells. * Denotes non-specific bands detected. $n = 3$ independent experiments. **e** *Ngfr* mRNAs were markedly increased in mouse articular chondrocytes (AC), E14.5 limb bud cells, and C2C12 cells, compared to C3H10T1/2, ATDC5, and CD45- BMSCs. Data are presented as mean values ± SD. $n = 3$ independent experiments. One-way ANOVA. **f, g** *NGFR* mRNAs were significantly higher in human osteoblastic cell lines (hFOB1.19 and Saos-2) than in human mesenchymal stem cells (hMSC), whereas *TrkA* levels are significantly lower than *NGFR* in all examined cells. Data are presented as mean values ± SD. $n = 3$ independent experiments. One-way ANOVA. **h, i** NGFR proteins decreased with the dedifferentiation of committed skeletal cells induced by repeated passages. $n = 3$. *$p < 0.05$, **$p < 0.01$, ****$p < 0.0001$. Source data are provided as a Source Data file.

groups, as shown by articular cartilage abrasion, synovial hyperplasia, osteophyte outgrowth and subchondral sclerosis, which are typical pathological features of OA (Supplementary Fig. 5a–e). Nevertheless, the progression of OA in the NGFR-deficient mice was indistinguishable from that in the control mice, as determined by Osteoarthritis Research Society International (OARSI) scoring and quantification of subchondral and ectopic bone formation (Supplementary Fig. 5b, e). Therefore, our results suggested that loss-of-function of NGFR in osteochondral cells induced in pre-adulthood may not have significant effects on joint pathophysiology, even when OA is inflicted in adulthood.

**Osteoarthritic joints require NGFR for bony remodeling and repair**

We recognized that tamoxifen administration at young ages may not target all of the osteochondral joint cells at a later stage because

osteochondral cells are incessantly differentiated from progenitor cells, and some of them are produced after tamoxifen injection, leaving the *Ngfr* gene untargeted. In fact, our IHC results confirmed that NGFR loss-of-function was not substantially achieved (only ~29% decrease) in the KO mice receiving tamoxifen at young ages (Supplementary Fig. 5f, g). In order to determine whether NGFR has an indispensable role during OA progression, we went on to perform the DMM surgery on a new cohort of 3-month-old mice and changed the method of tamoxifen administration. We tested weekly injection of tamoxifen on the *Acan-CreER; Ai9* reporter mice and found that this method efficiently targeted extensive types of osteochondral cells in the entire osteoarthritic joints, including those in articular cartilage, meniscus, subchondral bone, synovium and newly formed ectopic bone (Fig. 4a). It is notable that the targeted cells in the unoperated, healthy knee joints were primarily located in the cartilaginous tissues of articular cartilage and meniscus (Supplementary Fig. 6), suggesting that

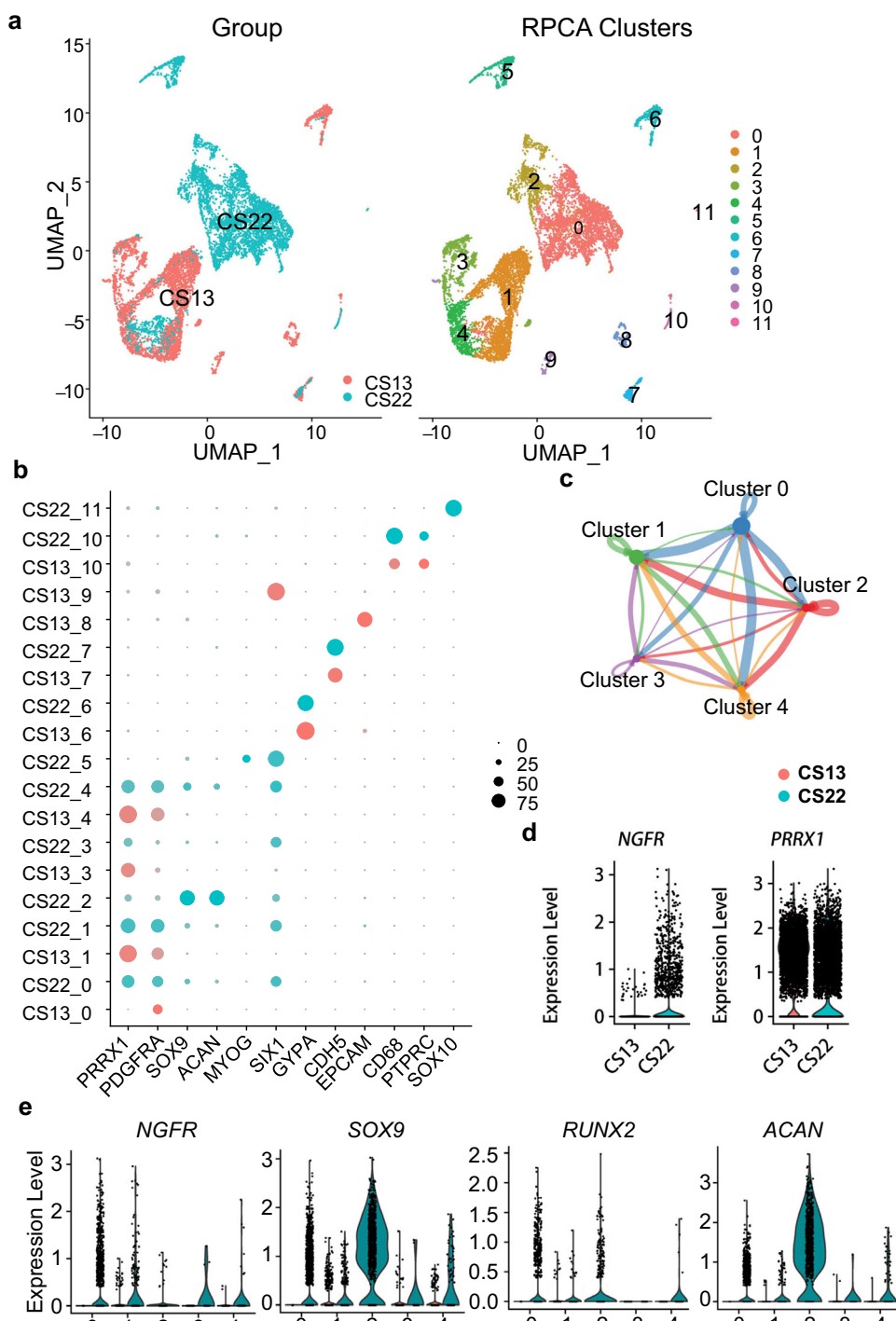

**Fig. 3 | NGFR is positively associated with osteochondral differentiation of human skeletal progenitor cells. a** The UMAP plot of 5 weeks post conception (Carnegie stage 13) human limb bud and 8 WPC (CS22) human long bone. **b** The dot plot showing expressions of representative genes in each cluster. Clusters 0–4 can be grouped as skeletal lineage cells based on their expression of *PRRX1* and *PDGFRA*. **c** The circle plot showing the aggregated cell–cell communication network and the total interaction strength (weights) between any two skeletal cell clusters. **d** Violin plots showing the cumulative expression of NGFR and PRRX1 in skeletal lineage cells of CS13 limb bud and CS22 long bone. **e** Characterization of the expression of NGFR, SOX9, RUNX2, and ACAN in each cluster of skeletal lineage cells, split by the groups of CS13 limb bud and CS22 long bone.

osteochondral remodeling is significantly more active in diseased joints than in healthy joints and confirming that *Acan-CreER* is a useful tool for studying bone remodeling in osteoarthritic joints. Thus, we injected the *Ngfr*$^{Acan-CreER}$ mice with tamoxifen weekly until we collected the knee joints 3 months after DMM. The IHC results confirmed an efficient ablation (90% decrease) of NGFR in the KO joints, whereas NGF remained induced in the osteoarthritic joints (Fig. 4b,

Supplementary Fig. 3b–d). Remarkably, both histology and micro-computed tomography (μCT) results demonstrated significant reductions in both ectopic and subchondral bones when *Ngfr* was continuously ablated (Fig. 4c–f). Quantitative analysis confirmed that the volume of ectopic bone/osteophytes was decreased by 36%, and medial subchondral bone volume fraction (BV/TV) was decreased by 21% in NGFR KO joints (Fig. 4e, f). It is notable that femoral trabecular

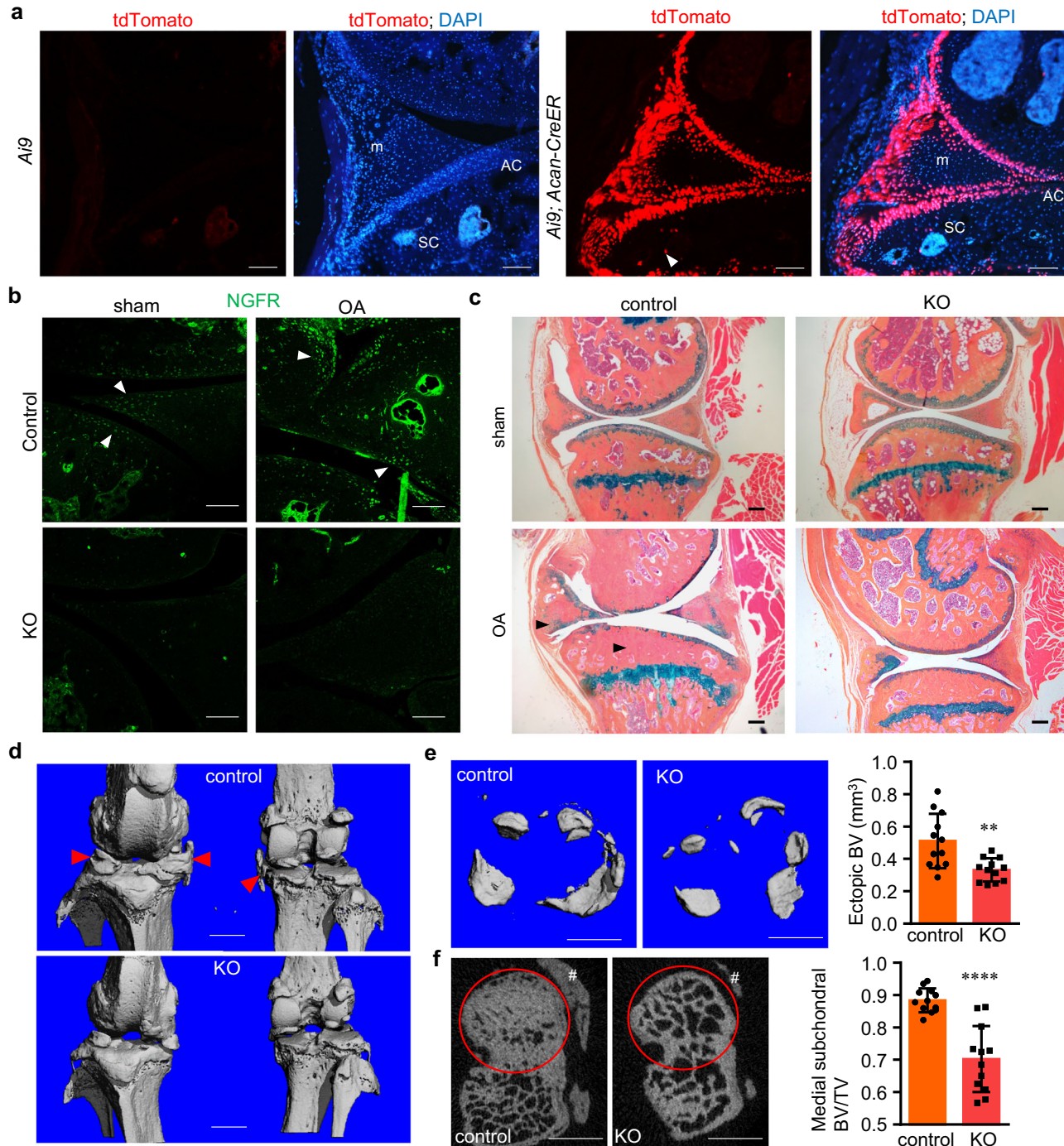

**Fig. 4 | NGFR deficiency substantially reduces subchondral bone thickening and ectopic bone formation in osteoarthritic joints. a** *Aggrecan-CreER* efficiently targets multiple osteochondral tissues in the osteoarthritic joints, including articular cartilage (AC), subchondral area (SC), and meniscus (m). Arrowhead, targeted cells in SC. Scale bar, 100 μm. *n* = 3 joints. **b** NGFR was induced in osteoarthritic joints. Both control (*Ngfr* floxed) and KO (*Ngfr^Acan-CreER*) mice were injected with tamoxifen weekly to ablate *Ngfr*, starting 10 days after the DMM surgery. Arrowheads: NGFR⁺ osteochondral cells. Scale bar, 100 μm. *n* = 5. **c** NGFR loss downregulated subchondral and ectopic bone formation (arrowheads) in osteoarthritic joints. Scale bar, 200 μm. *n* = 12 mice. **d** Representative μCT 3D

images of control and KO joints with OA. Arrowheads, ectopic bone. **e** Representative μCT 3D images of ectopic bone growth around cartilage, meniscus and synovium, as well as quantification of ectopic BV. Data are presented as mean values ± SD. *n* = 12 mice. Unpaired two-sided Student's *t*-test. **f** Representative μCT 2D images of subchondral bone and BV/TV quantification of subchondral bone underneath medial tibial plateau (inside the red circles). #, ectopic bone outside subchondral area. Scale bars: 1 mm. Data are presented as mean values ± SD. *n* = 12 mice. Unpaired two-sided Student's *t*-test. **p < 0.01, ****p < 0.0001. Source data are provided as a Source Data file.

bone volume fraction was not affected by NGFR deficiency in both female and male mice (Supplementary Fig. 7a, b), suggesting that NGFR loss-of-function mainly affected those skeletal cells close to the lesion, which is in line with our observation that sham joints did not

show obvious phenotypic changes. In addition, articular cartilage degradation and synovitis in the KO mice were similar to those in the control mice (Fig. 4c, Supplementary Fig. 7c, d). Thus, our data collectively suggested that NGFR deficiency at this stage predominantly

dysregulates skeletal cells recruited for bone remodeling in the joint. Intriguingly, bone loss or destruction was also observed in some OA patients receiving anti-NGF therapies, a significant adverse event[12]. Thus, we speculated that blockade of NGF-NGFR signaling may detrimentally downregulate bony remodeling of osteoarthritic joints, the spontaneous responses for joint repair.

## NGFR deficiency deteriorates inflammation-induced downregulation of osteogenesis

Because inflammatory cytokines, including tumor necrosis factor-α (TNF-α) and interleukin-1β can be detected in OA joint tissues[42,43], and inflammatory cytokines could lead to the inhibition of bone formation[44,45], we were interested to determine whether NGFR loss-of-function alters osteogenesis in skeletal cells in the context of inflammation. Thus, we cultured C2C12 cells with CRISPR-mediated deletion of *Ngfr* with osteogenic medium and stained the cells to examine the alkaline phosphatase (ALP) activity, a marker for osteogenesis. Our results showed that inflammatory cytokines caused a more striking reduction of ALP activity in NGFR KO cells (Fig. 5a). Therefore, our results suggested that NGFR may serve a resistant role to inflammation-induced impairment of osteogenesis in skeletal cells. Next, we performed western blot and found that the BMP-SMAD1 pathway was significantly downregulated in the NGFR KO cells under the circumstance of inflammatory cytokines, as displayed by the decrease of both phosphorylated and total SMAD1 proteins (Fig. 5b). Additionally, BMP-SMAD1 signaling was significantly upregulated in the CRISPRa cells treated with inflammatory cytokines, as evidenced by higher levels of SMAD1 and p-SMAD1 (Fig. 5c). The mRNA expressions of marker genes downstream of the BMP-SMAD1 pathway, including *Id1*, *Id2*, and the osteogenic master transcription factor *Runx2*, were all markedly reduced in the NGFR null cells (Fig. 5d–f). We also induced NGFR gain-of-function by CRISPRa in the mouse CD45⁻ BMSCs[20], which are early-stage progenitor cells and do not express significant amounts of NGFR (Fig. 2a), and found that NGFR significantly upregulated osteochondral genes, (Supplementary Fig. 8), further establishing the role of NGFR in promoting osteochondrogenesis.

To confirm whether the BMP-SMAD1 signaling was also repressed in the osteoarthritic joint of NGFR-deficient mice, we performed IHC to determine the levels of both SMAD1 and p-SMAD1. In the control knee joint subjected to the DMM surgery, both SMAD1 and p-SMAD1 were evidently upregulated in those areas expected to have ectopic ossification, such as hypertrophied synovium, calcified meniscus, and osteophytes (Fig. 5g–i), suggesting that OA generally stimulates anabolic responses in bone remodeling of osteoarthritic joints. Compared to the control joints, however, the NGFR KO joints showed significantly reduced levels of both SMAD1 and p-SMAD1, confirming that NGFR deficiency impairs the BMP signaling and thus downregulates bone formation that should have been induced to stabilize the diseased joints (Fig. 5g–i). Examination of RUNX2 also confirmed an impaired bone-forming response in osteoarthritic joints with NGFR deficiency (Fig. 5j, k). Together, our data demonstrated that NGFR deficiency impairs the bony stabilization of osteoarthritic joints.

## NGFR deters bone resorption and destruction in osteoarthritic joints

In order to investigate if NGFR perturbation has effects on osteoclast formation and bone resorption, we also checked the expression levels of RANKL and osteoprotegerin (OPG), the cytokine and its soluble decoy receptor, which are secreted by osteoblasts and other types of skeletal cells to regulate osteoclastogenesis. Interestingly, the *Rankl* mRNA level was significantly upregulated in the NGFR KO cells (Fig. 6a). Western blot also confirmed that the NGFR loss-of-function increases RANKL protein levels (Fig. 5b). Although *Opg* expression increased moderately in NGFR-null cells, the *Rankl/Opg* ratio was

negatively associated with NGFR under the treatment of inflammatory cytokines, suggesting that NGFR represses the induction of osteoclastogenesis by inflammation (Fig. 6b, c). Next, we treated control and NGFR KO cells with TNF and used supernatant medium from these cells to treat RAW 264.7 cells. Staining of tartrate-resistant acid phosphatase (TRAP), the enzyme responsible for bone resorption and established as an osteoclast marker, demonstrated that the medium conditioned by NGFR-deficient cells induced significantly stronger osteoclast differentiation than the medium conditioned by control cells (Fig. 6d, e), suggesting that NGFR loss-of-function may induce skeletal cells to produce higher levels of RANKL that enhance osteoclastogenesis.

Further, we performed TRAP staining on osteoarthritic NGFR-deficient joints to quantify osteoclast formation in subchondral and ectopic bone areas. Our results demonstrated that NGFR KO joints had more TRAP-positive cells than the control joints, particularly in subchondral areas, suggesting that NGFR deficiency in osteochondral cells significantly upregulated osteoclastogenesis (Fig. 6f, g). Moreover, we performed IHC to quantify the protein levels of RANKL and OPG in the joint tissues and found that NGFR deficiency resulted in a significant elevation of RANKL but not OPG proteins (Fig. 6h, i; Supplementary Fig. 9), suggesting that NGFR attenuated RANKL induction in osteoarthritic joints. Together, our results suggested a combinatory role of NGFR in both promoting bone formation and suppressing bone resorption to favor the bone-forming events, including ectopic bone/osteophyte outgrowth and subchondral sclerosis during OA pathogenesis.

## Negative correlation between NGFR and NF-κB activation

NGFR-intact and NGFR-KO joints appeared to display a stark contrast in bone remodeling when they developed OA, suggesting that OA may underlie abnormal bone turnover in NGFR-deficient joints. Our IHC results demonstrated that the majority of p65[46] translocated to the nucleus in the murine osteoarthritic joints, suggesting that OA generates an inflammatory milieu within the joints (Fig. 7a, b). Importantly, NGFR loss-of-function aggravated the inflammation as demonstrated by significantly increased nuclear localization of p65 in the osteoarthritic KO joints, especially in the joint tissues that underwent bone remodeling, such as the synovium and meniscus (Fig. 7c, d). Thus, our results suggested that NGFR negatively regulates NF-κB activation in murine osteoarthritic joints.

As our data showed that NGFR is induced in the cells treated with TNFα and in the murine osteoarthritic joints, we were also interested in investigating whether NGFR undergoes a perturbation in non-neuronal joint cells during human OA pathogenesis. The IHC results of NGFR demonstrated that the NGFR protein levels in different specimens of human osteoarthritic cartilage varied, but they were consistently higher than those in non-OA cartilage (Fig. 7e, f), suggesting that OA upregulates NGFR in human cartilage. We also performed IHC to detect p65 and found that the OA specimens exhibited greater frequencies of nuclear p65 than non-OA cartilage (Fig. 7g, h), confirming elevated NF-κB activity in human cartilage during OA. Interestingly, among different OA cartilage specimens, NGFR appeared to be negatively associated with the nuclear translocation of p65 in the examined samples (Fig. 7g). Together, our results suggested that the induction of NGFR could serve as a negative feedback mechanism in response to inflammation during human OA pathogenesis.

## NGF-NGFR signaling restricts inflammatory reaction through attenuating NF-κB activation

To examine how NGFR regulates the TNF/NF-κB signaling, we generated C2C12 cells with NGFR ablation by lentiviral transduction of 4 individual CRISPR sgRNAs, which was expected to exclude off-targeting effects. Western blot results showed that the sgRNAs effectively induced NGFR loss-of-function (Fig. 8a). More importantly, they

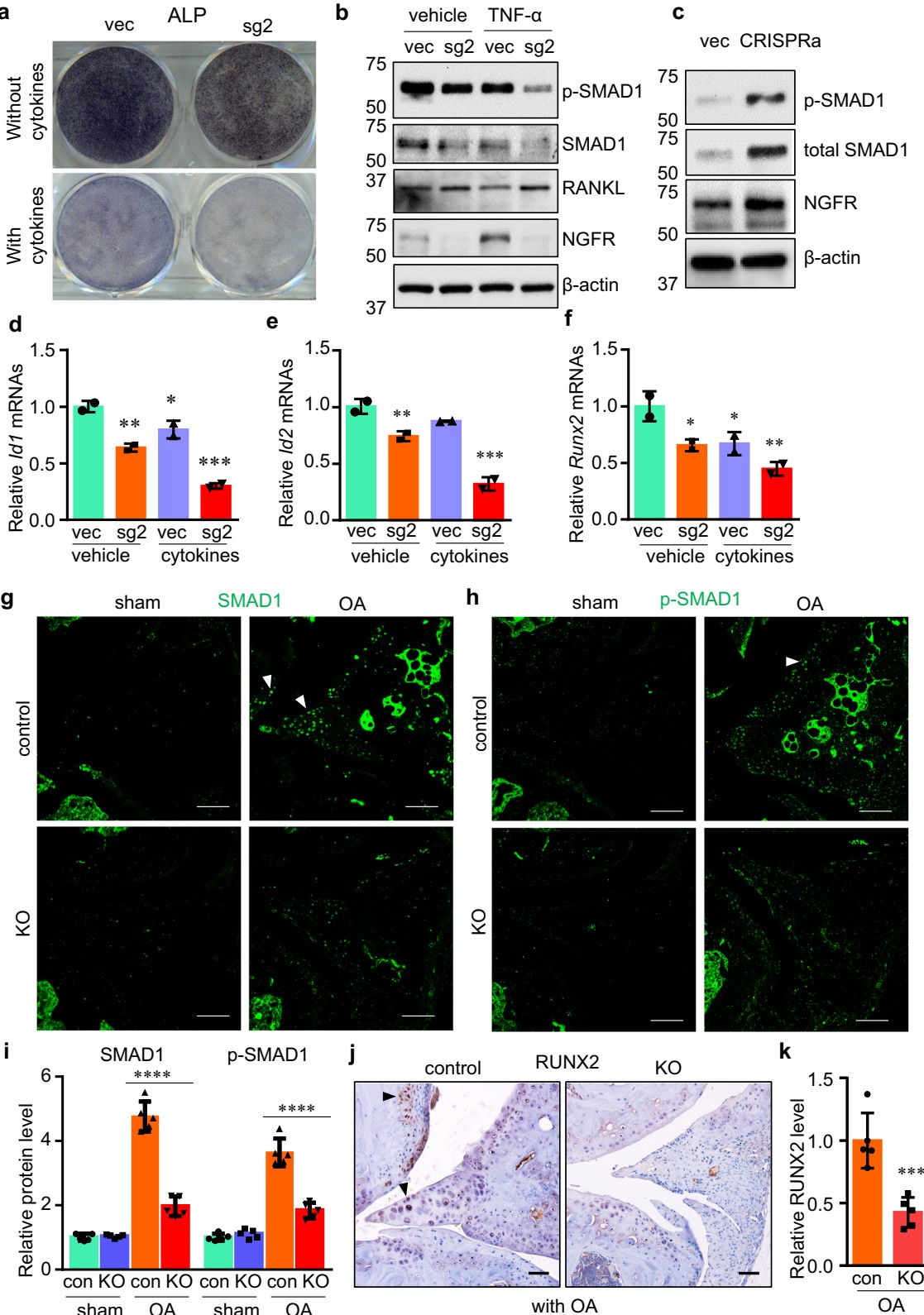

**Fig. 5 | Loss of NGFR impairs bone formation in the context of inflammation.** **a** NGFR ablation exacerbated the decrease of alkaline phosphatase activity induced by inflammatory cytokines. Vec, empty vector; sg2, CRISPR vector expressing single guide RNA 2 to mediate *Ngfr* deletion. *n* = 3. **b**, **c** NGFR perturbation altered SMAD1 and RANKL in TNF-α-treated cells. CRISPRa, CRISPR-mediated activation of *Ngfr*. *n* = 3 independent experiments. **d**–**f** NGFR deletion downregulated the marker gene expression in BMP-SMAD1 signaling. Data are presented as mean values ± SD.

*n* = 2 independent experiments. Two-way ANOVA. **g**–**i** OA-induced upregulations of SMAD1 and p-SMAD1 were significantly attenuated by NGFR deficiency in OA joints. Data are presented as mean values ± SD. *n* = 5 independent experiments. Two-way ANOVA. **j**, **k** NGFR deletion downregulated RUNX2 protein level in OA joints. Scale bar: 100 μm. Data are presented as mean values ± SD. *n* = 5 independent experiments. Unpaired two-sided Student's *t*-test. **p* < 0.05, ***p* < 0.01, ****p* < 0.001, *****p* < 0.0001. Source data are provided as a Source Data file.

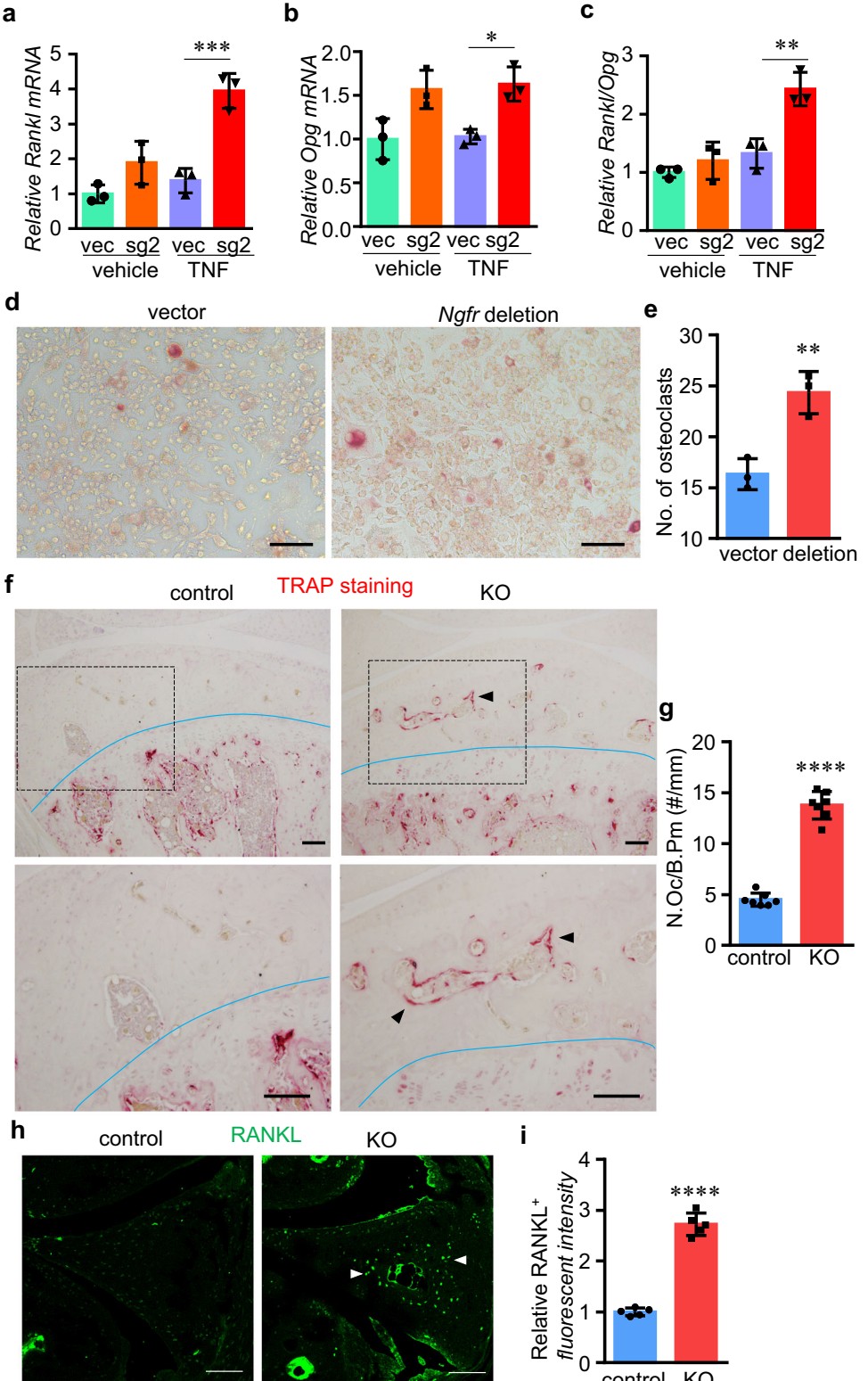

**Fig. 6 | NGFR loss-of-function further heightens inflammation-induced bone resorption. a–c** NGFR deletion induced the *Rankl/Opg* mRNA ratio in TNF-treated cells. Vec, empty vector; sg2, CRISPR vector expressing single guide RNA 2 to mediate Ngfr deletion. Data are presented as mean values ± SD. *n* = 3 independent experiments. Two-way ANOVA. **d**, **e** TRAP staining and quantification of RAW264.7 cells cultured with medium conditioned by empty vector-infected or CRISPR-mediated *Ngfr* deletion cells. Scale bar: 80 μm. *n* = 3 independent experiments. Data are presented as mean values ± SD. Unpaired two-sided Student's *t*-test. **f**, **g** TRAP staining and quantification of subchondral area in control and KO joints with OA. The lower edge of subchondral area is marked by the blue curves. The bottom panels are the enlargements of the upper images as indicated by the box, in order to show subchondral area in higher resolution. Arrowheads, positive TRAP staining in subchondral area. Scale bar: 100 μm. *n* = 7 independent experiments. Data are presented as mean values ± SD. Unpaired two-sided Student's *t*-test. **h**, **i** IHC of RANKL in the osteoarthritic joints with NGFR deficiency. Scale bar: 100 μm. Data are presented as mean values ± SD. *n* = 5 independent experiments. Unpaired two-sided Student's *t*-test. *\*p < 0.05, \*\*p < 0.01, \*\*\*p < 0.001, \*\*\*\*p < 0.0001.* Source data are provided as a Source Data file.

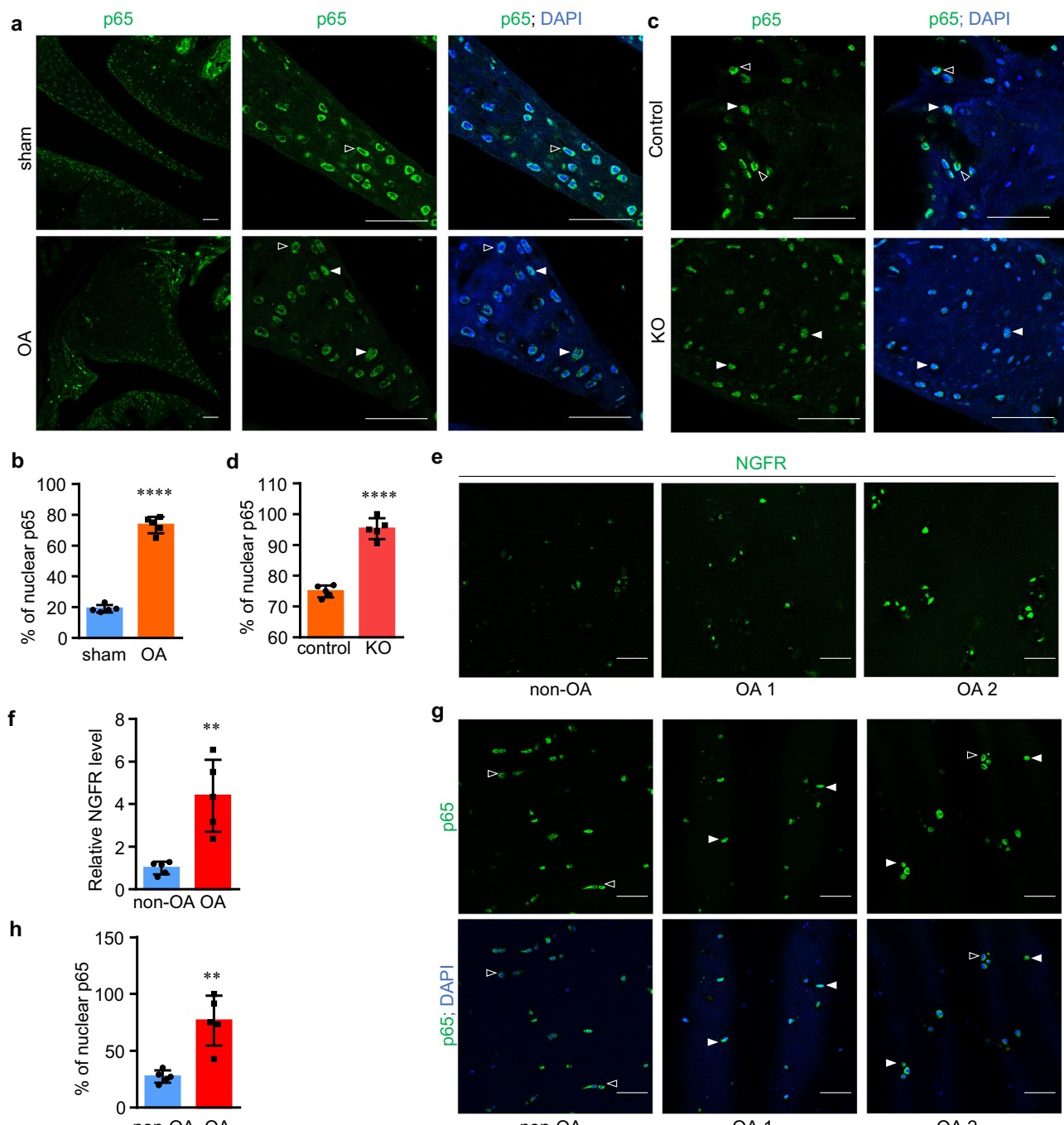

**Fig. 7 | NGFR attenuates NF-κB activation in skeletal tissues during OA. a, b** OA induces nuclear translocation of p65 in joint cells as shown by p65 IHC and quantification. Hollow arrowheads, cytoplasmic p65. Solid arrowheads, nuclear p65. Scale bar: 50 μm. Data are presented as mean values ± SD. $n = 5$ independent experiments. Unpaired two-sided Student's $t$-test. ****$p < 0.0001$. **c, d** Nuclear translocation of p65 in the joint cells induced by OA was exacerbated by NGFR deficiency. Hollow arrowheads, cytoplasmic p65. Solid arrowheads, nuclear p65. Scale bar: 50 μm. Data are presented as mean values ± SD. $n = 5$ independent experiments. Unpaired two-sided Student's $t$-test. ****$p < 0.0001$. **e, g** IHC of NGFR and p65 in human articular cartilage. OA 1 and OA 2 indicate different samples. **f, h** Quantification of IHC results of NGFR or nuclear p65. Data are presented as mean values ± SD. **$p < 0.01$. Scale bar: 50 μm. $n = 5$ independent experiments. Unpaired two-sided Student's $t$-test. Source data are provided as a Source Data file.

all consistently increased TNF-induced activation of NF-κB (Fig. 8a), suggesting that NGFR ablation exacerbates the inflammatory reactions. It is notable that the hyper-activation of NF-κB by NGFR deficiency occurred at the same time point when activation of TNF/ NF-κB reached the climax (Supplementary Fig. 10). To further explore whether NGF is also involved in this process, we treated the cells with TNF, NGF, or a combination of TNF and NGF. Interestingly, NGFR-deficient cells exhibited higher levels of p-p65 than NGFR-intact cells when

treated with NGF (Fig. 8b, Supplementary Fig. 11), suggesting that NGF plays an anti-inflammatory role that requires NGFR. Further, simultaneous treatment of TNF and NGF resulted in weaker phosphorylation of p65 than that of TNF alone in control cells, whereas NGFR-KO cells still showed strong p-p65, further confirming the potent TNF-blocking effects of NGF and NGFR (Fig. 8b). Remarkably, the total IKK level appeared to be elevated in the NGFR-deficient cells (Fig. 8b, Supplementary Fig. 10), pointing to a possible mechanistic clue by which

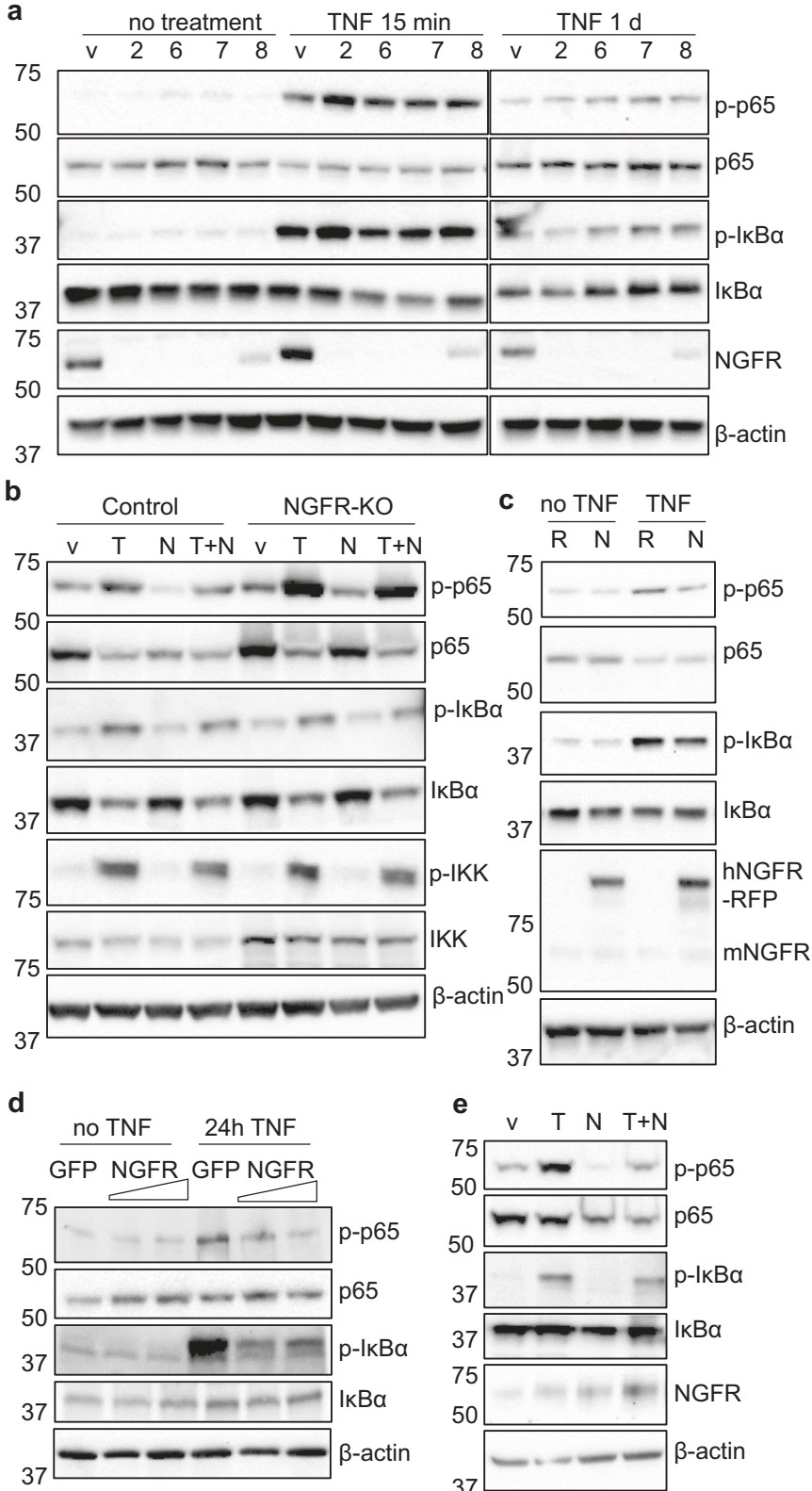

**Fig. 8 | NGF-NGFR restricts the TNF/NF-κB signaling. a** NGFR deletion enhanced TNF-induced phosphorylation of p65. v, empty vector; 2–8, different sgRNAs targeting *Ngfr*. *n* = 3 independent experiments. **b** NGF reduced TNF-induced p-p65, which required NGFR. v vehicle. T, TNF. N, NGF. T + N, TNF + NGF. Cells were treated with indicated growth factor/cytokine for 15 min. sgRNA #2 was used to generate NGFR KO. *n* = 3 independent experiments. **c** NGFR overexpression reduced p-p65 and p-IκBα in C2C12 cells treated with TNF for 24 h. R, RFP. N, hNGFR-RFP. *n* = 3 independent experiments. **d** NGFR overexpression decreased p-p65 and p-IκBα induced by TNF treatment in non-skeletal HEK293 cells. n = 3 independent experiments. **e** NGF treatment decreased NF-κB activation induced by TNF in HEK293 cells. *n* = 3 independent experiments. Source data are provided as a Source Data file.

NGFR modulates NF-κB signaling. To further test whether NGFR overexpression could decrease inflammatory response, we stably transfected C2C12 cells with NGFR-RFP or RFP and treated the cells with TNF. Our western blot results demonstrated that NGFR overexpression suppressed the phosphorylation of p65 and IκBα (Fig. 8c), suggesting that NGFR can restrict NF-κB activation induced by inflammatory cytokines. We were also interested in clarifying whether the inflammation-limiting role of NGFR could exist in non-skeletal cells. Thus, we transfected HEK293 cells with NGFR-RFP for NGFR overexpression and found that NGFR overexpression in HEK293 cells also restricted inflammation induced by TNF (Fig. 8d). In fact, when we treated HEK293 cells with TNF, NGF, or both, we found that NGF could decrease the NF-κB signaling activated by TNF (Fig. 8e), a result similar to what was observed in C2C12 cells. Together, our results suggested that this role of NGF-NGFR signaling could be widespread in various types of tissues and cells, which may reveal an example regarding how pain-associated molecules also modulate non-neuronal tissues.

## Discussion

A prominent phenotype observed from the NGFR deficient mice with surgically induced OA was the significant decrease of OA-related bone growth, such as subchondral sclerosis and ectopic bone/osteophyte outgrowth. This may be reminiscent of atrophic OA, which is suggested as a risk factor of RPOA. Generally, OA induces bone anabolism, while rheumatoid arthritis (RA) causes bone destruction due to significantly more severe inflammation[47–49]. Thus, the repair mechanism in joints affected by RA is more devastatingly impaired. In contrast, osteoarthritic joints usually have active bony remodeling, which is thought to be a necessary pathophysiologic response to counteract joint failure and to prevent more catastrophic consequences, especially when the trauma or degeneration of the joint is so severe that cartilage degradation is not reversible and cartilage cannot be repaired any more. Without proper stimulation of such a bone-anabolic response, arthritic joints could demonstrate accelerated bone loss and collapse. Thus, the phenotype in the NGFR KO mice could imply a mechanism underlying NGF inhibition-induced RPOA. Moreover, restriction of inflammation and upregulation of bone anabolism, as suggested by the role of NGFR, may implicate potential therapeutic options for RPOA or other bone-destructive rheumatic diseases.

The induction of NGFR in committed skeletal cells but not in early-stage stem cells suggests an important role of NGFR in skeletal biology. Intriguingly, the generally viable and fertile phenotype of NGFR null mice[27,28], which display abnormal reflexes or reduced innervation, seems to obscure whether NGFR has an indispensable function in skeletal development and homeostasis. Our analysis of the mice with conditional NGFR deficiency, which was induced at the age of two weeks, did not reveal an overt skeletal phenotype in both healthy and osteoarthritic joints. While this result may suggest that NGFR is not absolutely essential for skeletal cells, we speculated that it could represent a technical challenge in inducing genetic deficiency in those dynamically differentiated cells recruited to repair osteoarthritic joints. Since osteochondral cells are incessantly derived from progenitor cells, it is anticipated that the cells produced after tamoxifen injection would not have their *Ngfr* gene deleted. Therefore, we developed an alternative strategy of inducible gene KO by injecting tamoxifen weekly in order to induce *Ngfr* deletion in continuously derived osteochondral cells. Indeed, we found significant reductions of bony remodeling in NGFR-null joints (Fig. 4c–f). Thus, our results demonstrated that this strategy for inducible dynamic genetic deficiency could be useful, especially for those processes involving dynamic cell differentiation. Moreover, we anticipate there would be a lot more application scenarios for this strategy in addition to OA, like fracture healing in the skeletal field and many other chronic pathologic conditions. It is also recognized that tamoxifen treatment may increase bone formation in mice[50,51]. Thus, this side effect of tamoxifen as a reagent to induce conditional KO should be taken into account when designing the experiments and interpreting the results. With the inclusion of appropriate controls and considerations of the side effects, the strategy of long-term tamoxifen administration is expected to generate useful information when studying chronic or dynamic conditions.

To date, available studies of NGF signaling in skeletal biology were mostly focused on peripheral neuropathy that indirectly regulates skeletal tissues through vascular ingrowth or paracrine growth factors[52–54], while a recent report identified the role of NGF-NGFR signaling in coordinating skeletal cell migration[55]. Therefore, it remains understudied whether NGF directly regulates skeletal pathophysiology. Here our results established an important role of NGF-NGFR signaling in regulating non-neuronal skeletal cells and tissues. OA and other musculoskeletal diseases involve extensive tissue damage, which elicits nociceptive pain through activating inflammatory reactions[9,56]; therefore, pain acts as an alarm signal against noxious stimuli. The protective properties of pain could be further supplemented by the induction of pain-related factors, including NGF-NGFR in the damaged tissues, which may facilitate non-neuronal cells to restrict inflammation and facilitate tissue remodeling and repair as demonstrated by this study. Thus, it would be interesting to investigate if this inflammation-limiting role of NGF-NGFR can be extrapolated to numerous other painful musculoskeletal conditions, including multiple rheumatic diseases, bone fractures, and back pain, which would be instrumental to the successful development of safe, effective analgesics.

The lack of highly effective, low-risk pain therapy has caused inadequate treatment of OA pain. NGF antagonism had aroused hopes for its potential effectiveness and lower risk from misuse. However, recent clinical trials have suggested its negative effects on joint structure in OA patients[12,13]. Therefore, it would be significant if this therapy could be improved regarding its adverse effects of RPOA. As our results demonstrated that inhibition of NGF-NGFR leads to impaired bony remodeling of osteoarthritic joints, it could be promising to repurpose approved therapies targeting bone resorption or boosting bone formation to prevent or reduce the occurrence of RPOA. In addition, prescreening of certain biomarkers to evaluate inflammation, bone anabolism, or NGFR expression may help identify patients with higher RPOA risk. Together, a more in-depth understanding of the role of NGF-NGFR in skeletal cells will improve the development of therapeutic strategies for OA.

## Methods
### Human tissue acquisition
The study has been approved by the Institutional Review Board of Rush University Medical Center. Adult human knee joint cartilage tissues were obtained from the patients who underwent total knee arthroplasty because of OA or from human adult donors (cadavers) with no sign of cartilage degeneration within 24 h of death from donors via the Gift of Hope Organ and Tissue Donor Network (Elmhurst, IL), with approval by the local ethics committee and informed consent obtained from the families. No subjects have been recruited for the proposed project. The samples of both normal and OA tissues were de-identified prior to use in this study and were obtained with a code number in order to protect the individual's privacy. Thus, we as investigators, could not determine the patient's identity based on the coded sample.

### Animal studies
The animal protocol of this study has been approved by the Institutional Animal Care and Use Committee (IACUC) of the Rush University Medical Center, and all experimental methods and procedures were carried out in accordance with the approved guidelines. Mice were maintained in the Rush University Medical Center Comparative

Research Center. Animals were housed in the Micro-Isolator® system at 22 °C temperature, 45% humidity, and 12-h light/dark cycle and given food (2018 Teklad Global Rodent Diet) and water ad libitum. The *Ngfr* conditional knockout mice used in this study had heterozygous (a single copy) *Acan-CreER*[57] and homozygous floxed *Ngfr* (JAX Strain # 031162)[28]. Their Cre-negative littermates were used as the control mice. Sex-matched control and NGFR KO mice were used. We performed the DMM surgery on the right knee of the 3-month-old mice to induce OA[20,31]. Under anesthesia induced and maintained by isoflurane, the hind limbs were shaved and prepared for aseptic surgery. The right knee joint was exposed following a medial capsular incision and gentle lateral displacement of the extensor mechanism without transection of the patellar ligament. The medial meniscotibial ligament was dissected. After the replacement of the extensor and irrigation with saline to remove tissue debris, the medial capsular incision and the skin incision were closed. We monitored for general signs of stress and pain in the mice and found that the DMM surgery was well tolerated in the mice. Two strategies of tamoxifen injection (1 mg/10 g body weight, i.p. injection) were used respectively: 5-day injections when the mice are 15 days old; or weekly injections starting from 10 days after DMM until sample harvest. Both the control mice and the NGFR KO mice received tamoxifen injections. The DMM surgery induced post-traumatic osteoarthritis in both females and males and we did not observe significant differences regarding pathological changes in osteoarthritic joints between female and male mice after they were subjected to the DMM surgery. Thus, animal sex was not analyzed as a variable in this work. We also performed the DMM surgery on 6 male C57BL/6J mice to induce OA, which aimed to study the phenotypic and molecular changes during OA pathogenesis. We performed the sham operation on 6 male C57BL/6J mice by opening and exposing the structures of the knee and then closing the skin incision without manipulating joint tissues. *Acan-CreER* mice were crossed to *Ai9* (Rosa-CAG-LSL-tdTomato-WPRE, Jackson Strain # 007909)[58] to generate the *Acan-CreER* reporter mice, namely *Ai9;Acan-CreER*. Weekly injections of tamoxifen started 10 days after DMM on 3-month-old male *Ai9;Acan-CreER* or Ai9 male mice (*n* = 3) until the samples were collected two months later. The mice were euthanized under carbon dioxide.

## Cell isolation and culture

Mouse articular chondrocytes were isolated by dissecting articular cartilage from newborn pups and treating the cartilage with 3 mg/mL collagenase D in DMEM for 45 min at 37 °C and then in 0.5 mg/mL collagenase D overnight at 37 °C[37]. The cell suspension was filtered through a 70 µm cell strainer and cultured in DMEM with 10% FBS. CD45- mouse bone marrow stromal cells isolated from long bones from mouse long bones were maintained in our lab[59] and cultured in α-minimal essential medium (α-MEM) with 10% FBS. E14.5 Limb bud cells were isolated by dissecting limb buds from mouse embryos and incubating the tissue in 0.1% Trypsin/EDTA for 30 min at 37 °C. The limb buds were further dissociated by gentle pipetting, and the cell suspension was filtered through a 40-µM strainer and centrifuged. Then the cell pellets were resuspended and plated on culture dishes. To induce osteoblast differentiation, cells were cultured in α-MEM supplemented with 10% FBS, 10 nM dexamethasone, 50 µg/ml ascorbic acid, and 10 mM β-glycerophosphate. Alkaline phosphatase (ALP) was performed by fixing cells with 10% formalin and then incubated in 1-Step NBT/BCIP Substrate Solution (Thermo Fisher Scientific)[20]. We treated NGFR-null or control C2C12 cells with 1 ng/ml TNF for 32 h for the collection of supernatant medium. The medium was then supplemented with 50 ng/ml M-CSF and used to treat RAW264.7 cells seeded in the 96-well plates for a week. Then we performed TRAP staining[60], by incubating cells in TRAP staining solution (0.1 M sodium acetate, 50 mM sodium tartrate, pH 5.0, 0.2 mg/mL naphthol AS-MX phosphate, and 0.6 mg/mL Fast Red Violet LB Salt) at 37 °C for 5–30 min depending on microscopic observation.

## PCR, CRISPR, and plasmids

We extracted RNA with genomic DNA digested and designed quantitative PCR primers that span over a large (>2 kbp) intron for human or rodent *NGFR* and *TrkA* genes, which aims to avoid the interference of genomic DNA in the quantification of the mRNA levels. Quantitative RT-PCR was performed[20] using qScript cDNA Synthesis Kits (Quantabio), PerfeCTa SYBR® Green SuperMix Reaction Mixes (Quantabio), and the primer pairs for mouse *Ngfr* (5'- CCGCTGACAACCTCATTCCT-3' and 5'-TGTCGCTGTGCAGTTTCTCT-3'), rat *Ngfr* (5'-GGCCTTGT GGCCTATATTGC-3' and 5'-CTGTCGCTGTGCAGTTTCTC -3'), human *NGFR* (5'-CAGGACAAGCAGAACACCGT-3' and 5'-GGTGTGGACCGTG-TAATCCA-3'), mouse *TrkA* (5'-GCCTAACCATCGTGAAGAGTG-3' and 5'-CCAACGCATTGGAGGACAGAT-3'), rat *TrkA* (5'-ATGGGGACCTCAAC CGTTTC -3' and 5'-CAAAGGACCAGGAGCCACAT-3'), and human *TrkA* (5'-GGTACCAGCTCTCCAACACG-3' and 5'-CGCATGATGGCGTAGACC TC-3'), *Smad1* (5'- GCTTCGTGAAGGGTTGGGG-3' and 5'-CGGATGAAA-TAGGATTGTGGGG-3'), *Runx2* (5'-CAAGAAGGCTCTGGCGTTTA-3' and 5'-TGCAGCCTTAAATGACTCGG-3'), *Id1* (5'- CCTAGCTGTTCGCTGAAG GC-3' and 5'-CTCCGACAGACCAAGTACCAC-3'), *Id2* (5'- ATGAAAGCC TTCAGTCCGGTG-3' and 5'-AGCAGACTCATCGGGTCGT-3'), *Rankl* (5'-CAGCATCGCTCTGTTCCTGTA-3' and 5'-CTGCGTTTTCATGGAGTCT CA-3'), *Opg* (5'- ACCCAGAAACTGGTCATCAGC-3' and 5'-CTGCAATA-CACACACTCATCACT-3'), and *β-actin* (5'-GGCTGTATTCCCCTCCA TCG-3' and 5'-CCAGTTGGTAACAATGCCATGT-3'). To knockout *Ngfr* in the cultured cells, the vector of lentiCRISPR v2 (Addgene plasmid # 52961), a gift from Feng Zhang[61], was used for cloning of individual sgRNAs. The sgRNA sequences for mouse NGFR KO are listed as follows: sg2, CTCAGATGAAGCCAACCACG; sg6, ACAGGCATGTACACCC ACAG; sg7, TGGAGCAATAGACAGGAATG; and sg8, TATAGACTC CTTTACCCACG. For CRISPRa, we constructed Lenti-SpdCas9-VP64, a vector expressing dCas9-VP64 fusion protein based on pCMV-SpdCas9-VP64 (Addgene plasmid # 115794), a gift from Nadav Ahituv[62], and Lenti-EGFP-dual-gRNA, a vector constructed in our lab to express two gRNA scaffolds for SgCas9 and SaCas9 respectively. Multiple *Ngfr* promoter-targeting sgRNAs were cloned into Lenti-EGFP-dual-gRNA for generation of efficient CRISPRa, which include: sg1, GC AGTCAAGTGAGGCGTGAG; sg3, AGCATAACCGGAGGTGCCCT; sg4, GCGGTTCCGGAGGGGTTggg; and sg5, CCCACTGAGAAGCCACAGCG. NGFR-RFP was a gift from Moses Chao (Addgene plasmid # 24092).

## Western blot

Western blot analysis was performed[63], using the following antibodies: anti-NGFR, Cell Signaling, catalog # 8238S; anti-phospho-NF-κB p65 (Ser536), Cell Signaling, catalog #3033; anti-NF-κB p65 (D14E12), Cell Signaling, catalog #8242; anti-phospho-IκBα (Ser32), Cell Signaling, catalog # 2859; anti-IκBα, Novus Biologicals, catalog # NB100-56507; anti-p-SMAD1, Cell Signaling, catalog # 9516S; anti-SMAD1, abcam, catalog # ab63356; anti- Phospho-IKKα (Ser176)/IKKβ (Ser177), Cell Signaling, catalog #2078; anti-IKKβ, Cell Signaling, catalog #2678; anti-β-actin, Sigma-Aldrich, catalog # A5441; anti-RANKL, Novus Biologicals, Clone 12A668; anti-TrkA, Cell Signaling, catalog #2505; anti-Trk (pan) (A7H6R), Cell Signaling, catalog #92991. The dilutions of the antibodies are 1/1,000 except that of anti-β-actin is 1/10,000. Specifically, whole cell lysates for western blotting were extracted with lysis buffer containing 50 mM Tris (pH 7.6), 150 mM NaCl, 1 mM EDTA, 10% glycerol, and 0.5% NP-40 and protease inhibitor cocktail (Sigma). Protein samples were resolved by 10% SDS-PAGE gel and transferred to the nitrocellulose membranes, which were incubated with individual antibodies according to manufacturer's instruction and then incubated with a horseradish peroxidase-conjugated secondary antibody, Jackson ImmunoResearch, catalog # 111-035-003 (anti-rabbit) or 115-035-003 (anti-mouse). The blots were visualized by SuperSignal™ Western Blot Substrate (catalog # A45915) according to the manufacturer's instructions. Uncropped and unprocessed scans are provided in the Source Data file.

## Micro-CT, histology, and immunohistochemistry

We used a Scanco μCT35 scanner (Scanco Medical, Brüttisellen, Switzerland) with 55 kVp source and 145 μA current for formalin-fixed mouse legs with a resolution of 10 μm[64,65]. The scanned images from each group were evaluated at the same thresholds to allow 3-dimensional structural rendering of each sample. The start and end positions of the scan region are the mid-points of the femur and tibia respectively. For evaluation of the joint, the Volume of Interest (VOI) was chosen by contouring the scan regions that include the knee joint as well as partial femur and tibia. For evaluation of the subchondral bone, the VOI was chosen by drawing contours inside the medial side of the joint counterclockwise to include subchondral bone. For evaluation of ectopic bones, the VOI was chosen by drawing contours around the entire tissue counterclockwise and also drawing contours around the epiphysis clockwise. For evaluation of the trabecular bone, the VOI was selected by drawing contours inside the metaphyseal cortical bone counterclockwise and the femoral trabecular bone volume fraction (BV/TV) was measured. When the automatic contour function was used for multiple slices, we scrolled through each slice to check for accuracy. For the evaluation of the whole joint, subchondral bone, and trabecular bone, a lower threshold of 220 and an upper threshold of 1000 were chosen for all the groups. The 3D image of the joint, ectopic bone and the 2D image of subchondral bone were generated, and ectopic bone volume and medial subchondral bone volume fraction BV/TV were measured. For histology and immunohistochemistry (IHC), tissues were fixed in 10% formalin, decalcified, and embedded in paraffin. Serial sagittal sections of knee joints were cut every 3 μm from the medial compartments. The sections were stained with Alcian blue/hematoxylin & orange G (AB/H&OG) for histological analysis[66]. OARSI scoring was performed to evaluate knee joint AC destruction[20]. Specifically, both medial femoral condyle and medial tibial plateau were analyzed through three-level sections of the joints and the severity of OA is expressed as the summed scores for the entire joint. Synovitis was semi-quantified by the enlargement of the synovial lining cell layer according to a published scheme[67]. For TRAP staining, the sections were deparaffinized, rehydrated, and incubated in TRAP staining solution (0.1 M sodium acetate, 50 mM sodium tartrate, pH 5.0, 0.2 mg/mL naphthol AS-MX phosphate, and 0.6 mg/ml Fast Red Violet LB Salt) at 37 °C for 5-30 min depending on microscopic observation[60]. For immunohistochemistry[66], 3 μm paraffin sections were heated at 95 °C in Antigen Unmasking Solution (Vector Laboratories, Burlingame, CA, USA) for 10–15 minutes and then sequentially treated with 0.5% Triton X-100, Avidin/Biotin Blocking Kit (Invitrogen, Carlsbad, CA, USA). After blocking with 10% normal goat serum (Vector Laboratories, Burlingame, CA, USA) for 1 h, sections were treated with 1/100 anti-NGFR antibody (Cell Signaling, catalog # 8238S), 1/200 anti-RUNX2 antibody (MBL, catalog # D130-3), 1/200 NGF antibody (abcam, catalog # ab6199), 1/100 anti-p-SMAD1 (Cell Signaling, catalog # 9516S), 1/200 anti-SMAD1 (abcam, catalog # ab63356); 1/100 anti-p65 antibody (Cell Signaling, catalog #8242), 1/200 anti-RANKL (Novus Biologicals, Clone 12A668), 1/500 anti-osteoprotegerin (Novus Biologicals, Clone 98A1071), 1/200 anti-βIII-tubulin antibody (R&D Systems, catalog # MAB1195) overnight at 4 °C and incubated with secondary antibody conjugated to Alexa Fluor 488 (Thermo Fisher) for 30 min. Alternatively, we also used 1/400 secondary biotinylated goat anti-rabbit or anti-mouse antibody, biotinylated anti-streptavidin, and DyLight 488 streptavidin (Vector Laboratories) for fluorescence labeling and detection. For frozen sectioning of the knee joints of the Acan-CreER reporter mice, the dissected limbs were fixed in 4% Paraformaldehyde (PFA) for 3 days and decalcified in 14% EDTA for 2 weeks. The tissues were cryoprotected in 30% sucrose overnight before embedding in OCT and sectioning. Images of histology, IHC and frozen sections were captured using CellSens Imaging Software (Olympus) on an Olympus BX43 microscope, or a Zeiss LSM700 confocal microscope.

## Single cell RNA-seq data analysis

A scRNA-Seq dataset of human embryonic skeletogenesis was downloaded from Gene Expression Omnibus (GEO) under the accession number of (GSE143753)[38]. The expression matrix of two samples, CS13 limbbud and CS22 long bone respectively, were extracted and processed by the Seurat package (Version: 4.9.9.9060)[68]. We excluded low-quality cells in the analysis by filtering out those with a number of expressed genes less than 500 or greater than 6000, with a number of molecules detected less than 500 or greater than 40,000, or with a proportion of mitochondrial genes greater than 6%. After data normalization, 2000 highly variable features (genes) were identified and scaled. Then we performed principal component analysis (PCA) to reduce the dimensionality of the data. For an integrative analysis of the data from CS13 limbbud and CS22 long bone, we used the reciprocal PCA (RPCA) method to identify integration anchors and create an integrated data assay for clustering of the dataset and UMAP-based visualization. Specific marker genes were identified based on the procedures reported in the original publication of the dataset[38]. Gene expression data of cells and their cluster information were used to create a CellChat object for analyses of cell-cell communication networks according to the published protocol[69].

## Statistical analyses

All the data were expressed as mean ± s.d., as indicated in the figure legends. Statistical analyses (two-sided) were completed with Graph-Pad Prism. Unpaired Student's $t$-test (for two groups) and one-way or two-way ANOVA (for multiple groups) were used followed by the Tukey–Kramer test. $P < 0.05$ was considered statistically significant.

## Reporting summary

Further information on research design is available in the Nature Portfolio Reporting Summary linked to this article.

## Data availability

The data that support the findings of this study are available within the article, its Supplementary Information and source data files. Source data are provided with this paper. The dataset analyzed in this study from ref. 38 is available from the Gene Expression Omnibus (GEO) repository under the following accession number GSE143753. Source data are provided with this paper.

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

## Acknowledgements

We thank Jun Li for his technical expertise. This work was supported by Grants R01AR070222 and R01AR070222-04S1 of National Institutes of Health to J.H. and L.Z.

## Author contributions

J.H. conceived the study and wrote the manuscript. J.H. and L.Z. designed the experiments and analyzed data. J.H., L.Z., Y.L. and H.J. performed experiments. J.H. performed bioinformatic analysis. L.Z. revised the manuscript.

## Competing interests

The authors declare no competing interests.
