## [Peer Review File · Nature Communications]

REVIEWER COMMENTS

Reviewer #1 (Remarks to the Author):

The authors utilize conditional deletion of p75 in the DMM model to demonstrate a role for mesenchymal p75 in regulating inflammation, osteoanabolism, and regulation of osteoclasts in OA. The phenotype of conditional deletion mice is clearly demonstrated, and new information is gained from the study in addition to what is already known regarding NGF-TrkA signaling in OA. The overall focus is narrow, and the findings may be more suitable for specialty journal.

Figure 1 - A good degree of non-specific IHC staining is apparent in subfigures. Quantification could be more specific by histologic compartment / tissue. There is a heavy reliance on IHC analysis which could be augmented by other more quantitative methods. This comment applies to subsequent figures as well.

Figure 2 - Available single cell sequencing datasets could be used to strengthen the assertion that p75 is increased in cells with "committed osteochondral fates"

Since this is a new conditional KO animal it would be interesting to know if changes in bone mass or structure exist. Domains of reporter activity are expected to be diffuse within the joint, but this should be validated using a fluorescent reporter. Likewise, leakiness (or lack thereof) with or without DMM should be shown.

Reviewer #2 (Remarks to the Author):

Many thanks for the opportunity to review this interesting manuscript. The authors use multiple observational and interventional protocols in mouse and human studies, including in vivo and in vitro. Together their data indicate that NGFR is expressed by osteochondral cells, regulates bone turnover in OA (but not in early development), and that NGF and TNF pathways interact, such that their effects on bone turnover may be mutually dependent, or synergistic. The data provide insights (indirectly) into potential adverse and beneficial effects on NGF- and inflammation-suppression on bone/cartilage responses to joint injury/OA, and, intriguingly, offer possible explanations (and perhaps solutions, although this is not tested in the current study) to adverse events (RPOA2) observed during NGF-blockade pharmaceutical development programmes. The authors hypothesise that RPOA might be explained by blockade of NGF actions on NGFR in osteochondral cells. If correct, then these adverse effects might not be observed with TrkA inhibitors, nor if TNF were concurrently blocked alongside NGF. Both these approaches are subjects of pharmaceutical development, and, as such, the findings are of immediate practical interest.

In general the studies have been thoroughly and carefully executed and designed. The authors should be congratulated on not 'giving up' when they found that suppression of NGFR expression in early life did not affect OA. My main concerns relate to the interpretation (or at least phrasing of that interpretation) of the results.

Conclusions on the effects of pain are not justified by the current results: The authors do not measure

pain behaviour, and so any implications for pain are highly speculative. I suggest textual rephrasing to address this issue. Specific examples include: Abstract: 'direct role of pain in protecting organs'. This may be an overinterpretation of the results, in that the authors do not report pain as such, nor indeed any molecules at cellular sites that are likely to directly mediate pain. Given that NGFR is investigated on non-neuronal cells, and there is no evidence that NGF has an exclusive function in pain, terms such as 'NGF, the "painful" growth factor' may be misplaced in this manuscript. Also; 'a novel example regarding how pain modulates the pathophysiology of non-neuronal tissues', 'pain may play a more direct role through inducing and activating NGF-NGFR in non-neuronal cells' etc. Pain does not result in NGF or TNF expression, but rather NGF and TNF may cause pain. I agree that the current data indicate interactions between pain and inflammatory pathways, but not that pain itself can modulate those pathways. There is a separate literature and methodology on neurogenic inflammation that would need to be addressed to support such a statement, but I feel this would be a distraction from the important content of this paper.

Minor points:

'synovium calcification ... which replicated the pathological findings in OA patients' and ''joint tissues that underwent bone remodeling, such as synovium and meniscus'. Calcification or ossification of the synovium is not a characteristic feature of human OA. I think the authors might be referring to the formation of osteophytes. Calcification of osteophytes occurs within a cartilage matrix (endochondral ossification), and is contiguous with underlying bone and adjacent articular cartilage, rather than directly occurring within the synovium. Rat menisci are ossified, but human menisci are not, so the relevance of menisci to this work is unclear.

Which sex mice were used in the experiments? This is particularly relevant given that tamoxifen was used, but also because of sex-specific influences on OA.

Please avoid overinterpretation of this as a model of RPOA. 'bone volume of osteophytes around meniscus and synovium was decreased by 36%, and medial subchondral bone volume fraction (BV/TV) was decreased by 21%' 'reminiscent of RPOA'. The cited reference describes RPOA as 'RPOA1 was defined as a significant decline in joint space width (JSW) ≥ 2 mm (predicated on optimal joint positioning) within approximately 1 year, without gross structural failure (19). RPOA2 was defined as abnormal bone loss or destruction, including limited or total collapse of ≥ 1 subchondral surface, that is not normally present in conventional end-stage OA'. Please either rephrase, or provide citations to support the statement that reduced osteophytosis and subchondral bone volume are characteristics of RPOA type 2. I am unaware of this literature. Certainly the model does not reflect RPOA type 1 in that cartilage damage in the model is similar to that without NGFR KO.

Relevance of TNF to OA structural change: The manuscript's thesis might be clearer if the authors could succinctly summarise the current state of knowledge for the role of inflammatory cytokines, specifically TNF which they use in in vitro experiments, in structural changes of OA. This is important regarding the translation relevance of the mouse model, which might have a more inflammatory phenotype than has human OA (although this is debated in the literature, and inflammation was not directly reported in the current experiments). My own perspective is that inflammation is common in OA, it is associated with

structural OA severity, but that specific blockade of TNF has not been definitively shown to reduce either structural damage nor pain in human OA. It is relevant that AstraZeneca have been developing in Phase III trials a dual NGF/TNF inhibitor. The authors appropriately discuss relevance to inflammatory arthritis. Their findings do beg the question as to whether the observed effects of NGFR KO on bone structure would be prevented by TNF inhibition.

The manuscript would benefit from grammatical and textual review (e.g. 'promote the growth of nervous system', 'massively abundant' might not be appropriate when discussing molecules, 'weakening of osteogenesis', 'more dramatic reduction' and 'dramatic decrease of OA-related bone growth' (in what way does this relate to drama??), 'rampant opioid crisis', 'vital role of NGF-NGFR signaling in regulating non-neuronal skeletal'; I'm not convinced this is 'vital' when referring to a non-fatal condition. 'reductions of ... subchondral bones'. 'impairs bony stabilization of osteoarthritic joints'. I'm uncomfortable with the word 'stabilization' here, in that the authors did not assess mechanical stability, commonly deficient in OA joints. There is a literature (without strong evidence) suggesting teleologically that osteophytes are adaptive changes in OA to help stabilise the joint. To avoid confusion, might alternative phrasing be used? Do the authors mean 'bone homeostasis', 'bone turnover'. 'Atrophic OA, which is regarded as a risk factor of RPOA'; please give a citation with evidence that supports the suggestion that atrophic OA is a risk factor for RPOA. I understand that this has been raised as a 'concern', but am unaware of evidence that it was correct. 'Has been suggested might be a risk factor' might be more accurate phrasing (supported by risk-mitigation strategies used in the tenazumab post-FDA-hold studies).

A possibly important implication of the current findings would be that if they are relevant to the pathogenesis of RPOA, this adverse effect might not be expected with specific inhibitors of TrkA.

Reviewer #3 (Remarks to the Author):

This study by Zhao et al. investigates the role of NGF receptor on bone remodeling and OA progression. They didn't observe any effect of NGFR knockout on joint development or OA progression with early tamoxifen injection, but saw notable differences when tamoxifen was administered weekly in adult mice. Additional in vitro studies established specific mechanisms and expression. Altogether these studies present novel data implicating an understudied pathway in bone metabolism and joint degeneration. However, there are several issues that need to be addressed before this manuscript is considered for publication in Nature Communications.

Major Issues:

The primary issue with this study is an overall lack of detail in the Methods section. For example, much more information is needed about microCT, in particular volume of interest, contouring, thresholding, outcomes, etc.

Why were sagittal sections used? Usually for DMM studies frontal sections are used. Figure 1 clearly

shows frontal sections, while Figure 3 shows sagittal sections. Why are these different? Are these even from the same study?

Tamoxifen injections could considerably affect bone remodeling. Were control mice also injected with tamoxifen, or just the KO mice? Please state this in the Methods. Why were injections started 10 days after DMM?

Did you perform sham surgeries as control for DMM?

Need better description of study design, animal numbers, animal sex, time points, experimental groups, etc.

Minor Issues:

Abstract: what does “stabilization of osteoarthritic joints” mean?

“This study uncovers a direct role of pain in protecting organs...” Pain is not measured in this study, therefore it is not possible to make this association. Although NGF is most commonly associated with pain, this relationship would have to be specifically investigated before this conclusion can be reached.

Introduction:

“Subchondral sclerosis is an adaptive response of the diseased joint to repair the damage caused by the overload”. I don't know if I agree with this sentence. Please reword, remove, or provide supporting references. The next sentence about osteophytes is also reductive.

Language/grammar is generally good, but could use some editing in some places.

Need to define all acronyms at their first use.

How/why did the authors choose the cell lines that you did?

Please be consistent with using TrkA vs. Ntrk1.

All graphs should show individual data points

Where are the results on OARSI score? This is a primary outcome, but the data is not shown. Was grading of synovitis performed? It doesn't appear so, but this should definitely be done.

REVIEWER COMMENTS

Reviewer #1 (Remarks to the Author):

The authors utilize conditional deletion of p75 in the DMM model to demonstrate a role for mesenchymal p75 in regulating inflammation, osteoanabolism, and regulation of osteoclasts in OA. The phenotype of conditional deletion mice is clearly demonstrated, and new information is gained from the study in addition to what is already known regarding NGF-TrkA signaling in OA. The overall focus is narrow, and the findings may be more suitable for specialty journal.

Figure 1 - A good degree of non-specific IHC staining is apparent in subfigures. Quantification could be more specific by histologic compartment / tissue. There is a heavy reliance on IHC analysis which could be augmented by other more quantitative methods. This comment applies to subsequent figures as well.

Response: We thank the reviewer for the critical comment. For the IHC staining data listed in the revised Fig. 1f,g and Fig. 5j, we have repeated the experiments and generated new images for them. In addition, we would like to explain that the brightness of most of our fluorescent IHC data had been adjusted up in the last version of the manuscript compared to the original images captured under the microscope, because we were suggested to do so to increase the visibility of the structures. As the reviewer has pointed out this issue, we have adjusted the brightness back to its original status in the revised manuscript. To show the structure of the tissues in these IHC images, we have now included the DAPI staining images in the supplementary data. With these revisions, more specific IHC staining signals as well as their histologic compartment information could be more evident.

Figure 2 - Available single cell sequencing datasets could be used to strengthen the assertion that p75 is increased in cells with "committed osteochondral fates"

Response: We appreciate this very constructive comment from the reviewer. As suggested by the reviewer, we have included an analysis of a published single cell sequencing dataset of human embryonic skeletogenesis in the revised manuscript (new Fig. 3), which shows that NGFR is increased in skeletal cells with more committed osteochondral fates.

Since this is a new conditional KO animal it would be interesting to know if changes in bone mass or structure exist. Domains of reporter activity are expected to be diffuse within the joint, but this should be validated using a fluorescent reporter. Likewise, leakiness (or lack thereof) with or without DMM should be shown.

Response: As the reviewer suggested, we have included the data of trabecular bone mass in the KO mice (new Supplementary Fig. 7), which demonstrated that the trabecular bone mass was not significantly altered in the KO mice. Also, we agree that the reviewer's comments about the reporter activity is very important, thus we had been working on generating the Aggrecan-CreER; Ai9(tdTomato) mice, and started to induce osteoarthritis in the mice just days before we received the review comments. To meet the requirement of submitting the revision in 3 months, we injected the mice with tamoxifen for 8 weeks and decalcified/sectioned the joints for fluorescent microscopy. Our results (new Figure 4A and Supplementary Fig. 6) clearly demonstrated that Aggrecan-CreER could target multiple types of osteochondral cells in the osteoarthritic joints, which is indeed "diffuse within the joint". In the healthy joint, the reporter activity is mainly located in the cartilaginous tissue, suggesting that bone remodeling in the

healthy joints was not as active as that in the diseased joints. The Cre-negative control also showed that the reporter is specific for skeletal tissues.

Reviewer #2 (Remarks to the Author):

Many thanks for the opportunity to review this interesting manuscript. The authors use multiple observational and interventional protocols in mouse and human studies, including in vivo and in vitro. Together their data indicate that NGFR is expressed by osteochondral cells, regulates bone turnover in OA (but not in early development), and that NGF and TNF pathways interact, such that their effects on bone turnover may be mutually dependent, or synergistic. The data provide insights (indirectly) into potential adverse and beneficial effects on NGF- and inflammation-suppression on bone/cartilage responses to joint injury/OA, and, intriguingly, offer possible explanations (and perhaps solutions, although this is not tested in the current study) to adverse events (RPOA2) observed during NGF-blockade pharmaceutical development programmes. The authors hypothesise that RPOA might be explained by blockade of NGF actions on NGFR in osteochondral cells. If correct, then these adverse effects might not be observed with TrkA inhibitors, nor if TNF were concurrently blocked alongside NGF. Both these approaches are subjects of pharmaceutical development, and, as such, the findings are of immediate practical interest.

In general the studies have been thoroughly and carefully executed and designed. The authors should be congratulated on not 'giving up' when they found that suppression of NGFR expression in early life did not affect OA. My main concerns relate to the interpretation (or at least phrasing of that interpretation) of the results.

Conclusions on the effects of pain are not justified by the current results: The authors do not measure pain behaviour, and so any implications for pain are highly speculative. I suggest textual rephrasing to address this issue. Specific examples include: Abstract: 'direct role of pain in protecting organs'. This may be an overinterpretation of the results, in that the authors do not report pain as such, nor indeed any molecules at cellular sites that are likely to directly mediate pain. Given that NGFR is investigated on non-neuronal cells, and there is no evidence that NGF has an exclusive function in pain, terms such as 'NGF, the "painful" growth factor' may be misplaced in this manuscript. Also; 'a novel example regarding how pain modulates the pathophysiology of non-neuronal tissues', 'pain may play a more direct role through inducing and activating NGF-NGFR in non-neuronal cells' etc. Pain does not result in NGF or TNF expression, but rather NGF and TNF may cause pain. I agree that the current data indicate interactions between pain and inflammatory pathways, but not that pain itself can modulate those pathways. There is a separate literature and methodology on neurogenic inflammation that would need to be addressed to support such a statement, but I feel this would be a distraction from the important content of this paper.

Response: We thank the reviewer for the positive and constructive comments. We agree with the reviewer that some of our interpretations of the results may need to be validated by additional work that could be a distraction from this paper, thus we have rewritten all these sentences and removed those overinterpretation in the revised manuscript. Please see the revisions highlighted in red color throughout the manuscript.

Minor points:

'synovium calcification ... which replicated the pathological findings in OA patients' and ''joint tissues that underwent bone remodeling, such as synovium and meniscus'. Calcification or ossification of the synovium is not a characteristic feature of human OA. I think the authors

might be referring to the formation of osteophytes. Calcification of osteophytes occurs within a cartilage matrix (endochondral ossification), and is contiguous with underlying bone and adjacent articular cartilage, rather than directly occurring within the synovium. Rat menisci are ossified, but human menisci are not, so the relevance of menisci to this work is unclear.

Response: As the reviewer commented, we were referring to the formation of osteophytes. We also agree with the reviewer that rodents (mice and rats) are different from humans regarding osteophyte formation and meniscus ossification, which is actually an important issue to consider when comparing OA progression between humans and rodents. Thus, we have revised the sentence “replicated the pathological findings in OA patients” to “underscores synovial inflammation and bony remodeling in OA progression”.

Which sex mice were used in the experiments? This is particularly relevant given that tamoxifen was used, but also because of sex-specific influences on OA.

Response: We have used both sexes in the experiments. We have added the sentence “Sex-matched control and NGFR KO mice were used” in the methods and added related information in the Supplementary Figure 7 .

Please avoid overinterpretation of this as a model of RPOA. `bone volume of osteophytes around meniscus and synovium was decreased by 36%, and medial subchondral bone volume fraction (BV/TV) was decreased by 21%` `reminiscent of RPOA`. The cited reference describes RPOA as `RPOA1 was defined as a significant decline in joint space width (JSW) ≥ 2 mm (predicated on optimal joint positioning) within approximately 1 year, without gross structural failure (19). RPOA2 was defined as abnormal bone loss or destruction, including limited or total collapse of ≥ 1 subchondral surface, that is not normally present in conventional end-stage OA`. Please either rephrase, or provide citations to support the statement that reduced osteophytosis and subchondral bone volume are characteristics of RPOA type 2. I am unaware of this literature. Certainly the model does not reflect RPOA type 1 in that cartilage damage in the model is similar to that without NGFR KO.

Response: As the reviewer suggested, we have rewritten this sentence, which now reads “bone loss or destruction was also observed in some OA patients receiving anti-NGF therapies” . We cited two papers (Ref. 14, 15) which discussed subchondral insufficiency fracture and atrophic OA as potential risk factors for RPOA (which could be “general” RPOA and may not be related to anti-NGF). Nevertheless, we agree that our results did not reflect RPOA1 (although we would like to suspect that some RPOA1 may advance to RPOA2 if the clinical trial was not discontinued) and that there is no paper about reduced osteophytosis and subchondral bone volume in anti-NGF receiving patients. Thus, we have made the revisions to avoid the overinterpretation.

Relevance of TNF to OA structural change: The manuscript’s thesis might be clearer if the authors could succinctly summarise the current state of knowledge for the role of inflammatory cytokines, specifically TNF which they use in in vitro experiments, in structural changes of OA. This is important regarding the translation relevance of the mouse model, which might have a more inflammatory phenotype than has human OA (although this is debated in the literature, and inflammation was not directly reported in the current experiments). My own perspective is that inflammation is common in OA, it is associated with structural OA severity, but that specific blockade of TNF has not been definitively shown to reduce either structural damage nor pain in human OA. It is relevant that AstraZeneca have been developing in Phase III trials a dual

NGF/TNF inhibitor. The authors appropriately discuss relevance to inflammatory arthritis. Their findings do beg the question as to whether the observed effects of NGFR KO on bone structure would be prevented by TNF inhibition.

Response: As the reviewer suggested, we have added a brief introduction in the Results of the revised manuscript about the rationale we were testing TNF in our in vitro experiments. We agree that it will be interesting to investigate if TNF inhibition or inflammation suppression could prevent the effects of NGFR KO. Although TNF inhibition did not meet with therapeutic success in OA patients like it does in RA patients, it could be useful for the treatment of specific subgroups of RPOA.

The manuscript would benefit from grammatical and textual review (e.g. ‘promote the growth of nervous system’. ‘massively abundant’ might not be appropriate when discussing molecules, ‘weakening of osteogenesis’, ‘more dramatic reduction’ and ‘dramatic decrease of OA-related bone growth’ (in what way does this relate to drama??), ‘rampant opioid crisis’. ‘vital role of NGF-NGFR signaling in regulating non-neuronal skeletal’; I’m not convinced this is ‘vital’ when referring to a non-fatal condition. ‘reductions of ... subchondral bones’. ‘impairs bony stabilization of osteoarthritic joints’. I’m uncomfortable with the word ‘stabilization’ here, in that the authors did not assess mechanical stability, commonly deficient in OA joints. There is a literature (without strong evidence) suggesting teleologically that osteophytes are adaptive changes in OA to help stabilise the joint. To avoid confusion, might alternative phrasing be used? Do the authors mean ‘bone homeostasis’, ‘bone turnover’. ‘Atrophic OA, which is regarded as a risk factor of RPOA’; please give a citation with evidence that supports the suggestion that atrophic OA is a risk factor for RPOA. I understand that this has been raised as a ‘concern’, but am unaware of evidence that it was correct. ‘Has been suggested might be a risk factor’ might be more accurate phrasing (supported by risk-mitigation strategies used in the tenazumab post-FDA-hold studies).

Response: As the reviewer suggested, we have revised the sentences (e.g. “promote the growth of nervous system” is changed to “be essential for the growth and survival of neurons”, “massively abundant” changed to “abundant”, “weakening” to “downregulation”, “dramatic” or “dramatically” replaced with “significant”, or “significantly” or “markedly”, “rampant” deleted, “vital” to “important”, “stabilization” changed to “repair” or deleted, etc). As the reviewer suggested, we have changed “regarded as a risk factor” to “suggested as a risk factor” and provided two new references (Ref. 14, 15) that may support the suggestion that atrophic OA is a risk factor for “general” RPOA (Ref. 15 is a systematic review published in 2002 and suggests a faster progression of hip OA when there is atrophic bone response).

A possibly important implication of the current findings would be that if they are relevant to the pathogenesis of RPOA, this adverse effect might not be expected with specific inhibitors of TrkA.

Response: We agree with the reviewer on this hypothesis. If there are highly specific, effective TrkA inhibitors (we are not experts for a discussion if those tested small molecule TrkA inhibitors are comparable to anti-NGF in terms of pharmacological properties), the adverse effect might be avoided.

Reviewer #3 (Remarks to the Author):

This study by Zhao et al. investigates the role of NGF receptor on bone remodeling and OA

progression. They didn't observe any effect of NGFR knockout on joint development or OA progression with early tamoxifen injection, but saw notable differences when tamoxifen was administered weekly in adult mice. Additional in vitro studies established specific mechanisms and expression. Altogether these studies present novel data implicating an understudied pathway in bone metabolism and joint degeneration. However, there are several issues that need to be addressed before this manuscript is considered for publication in Nature Communications.

Major Issues:

The primary issue with this study is an overall lack of detail in the Methods section. For example, much more information is needed about microCT, in particular volume of interest, contouring, thresholding, outcomes, etc.

Response: we thank the reviewer for the constructive comments. As the reviewer suggested, we have added more details in the Methods section, including the information about microCT, which reads as follows: We used a Scanco μ CT35 scanner (Scanco Medical, Brüttsellen, Switzerland) with 55 kVp source and 145 μ Amp current for formalin-fixed mouse legs with a resolution of 10 μ m as previously described. The scanned images from each group were evaluated at the same thresholds to allow 3-dimensional structural rendering of each sample. The start and end positions of the scan region are the mid-points of the femur and tibia respectively. For evaluation of the joint, the Volume of Interest (VOI) was chosen by contouring the scan regions that include the knee joint as well as partial femur and tibia. For evaluation of the subchondral bone, the VOI was chosen by drawing contours inside the medial side of the joint counterclockwise to include subchondral bone. For evaluation of ectopic bones, the VOI was chosen by drawing contours around the entire tissue counterclockwise and also drawing contours around the epiphysis clockwise. For evaluation of the trabecular bone, the VOI was selected by drawing contours inside the metaphyseal cortical bone counterclockwise and the femoral trabecular bone volume fraction (BV/TV) were measured. When the automatic contour function was used for multiple slices, we scrolled through each slice to check for accuracy. For evaluation of the whole joint, subchondral bone, and trabecular bone, a lower threshold of 220 and the upper threshold of 1000 were chosen for all the groups. The 3D image of the joint, ectopic bone, and the 2D image of subchondral bone were generated, and ectopic bone volume and medial subchondral bone volume fraction BV/TV were measured.

Why were sagittal sections used? Usually for DMM studies frontal sections are used. Figure 1 clearly shows frontal sections, while Figure 3 shows sagittal sections. Why are these different? Are these even from the same study?

Response: We performed the DMM surgery to induce OA, which was operated on the medial side and expected to have more prominent effects on the medial side. Specifically, we observed more ectopic bone formation in the frontal and rear sides of the joints, so a sagittal section would be able to reveal these prominent changes. Therefore we chose to use sagittal sections which can display more comprehensive pathological changes on the medial side. We also observed that sagittal sections are also used by some colleagues in the OA field, for examples, please see these publications on *Ann Rheum Dis*, a journal with emphasis on arthritis in 2023: Courties A, et al., Ohzono et al., and Knights et al.

We also agree with the reviewer that frontal sections can be useful. In our experiences we found that frontal sections were less prone to fall off the slides during heat induced antigen retrieval for immunohistochemistry. It can also provide a comprehensive view of both medial and lateral sides. Thus, we prepared frontal sections for the study presented in Figure 1. The studies

presented in Figure 1 and new Figure 4 used two different strains, which are WT and NGFR knockout mice respectively, but both of them were performed in our lab in recent years to identify a potentially important role of the NGF-NGFR pathway.

Tamoxifen injections could considerably affect bone remodeling. Were control mice also injected with tamoxifen, or just the KO mice? Please state this in the Methods. Why were injections started 10 days after DMM?

Response: We agree with the reviewer that tamoxifen injections could affect bone remodeling; thus we included both the control mice and the KO mice in each weekly injection of tamoxifen. As suggested, we have added a sentence “Both the control mice and the NGFR KO mice received tamoxifen injection.” in the revised Methods. The reason why injections started 10 days after DMM is because we had found that the first week after DMM has very active inflammatory responses, but bone remodeling appeared to be evident later. Thus, we suspected that injections started 10 days after DMM can still have effects on joint remodeling. The other reason we did not inject tamoxifen earlier is because both the DMM surgery and tamoxifen injection are stressful to the mice, so we performed injections 10 days after DMM.

Did you perform sham surgeries as control for DMM?

Response: We have performed sham surgeries as control for DMM in the experiments as labeled in the Figures.

Need better description of study design, animal numbers, animal sex, time points, experimental groups, etc.

Response: As suggested, we have added the information in the Methods and also described important information in the Results, Figure Legends and Supplementary Figure Legends.

Minor Issues:

Abstract: what does “stabilization of osteoarthritic joints” mean?

“This study uncovers a direct role of pain in protecting organs...” Pain is not measured in this study, therefore it is not possible to make this association. Although NGF is most commonly associated with pain, this relationship would have to be specifically investigated before this conclusion can be reached.

Response: As two reviewers have commented on “stabilization of osteoarthritic joints” , we feel this may not be the best way to describe bone remodeling in the joint, thus we have deleted this word in our revised manuscript. Similarly, we also revised the sentence “This study uncovers a direct role of pain in protecting organs...” to “this study uncovers a role of NGFR in limiting inflammation for repair of diseased skeletal tissues”.

Introduction:

“Subchondral sclerosis is an adaptive response of the diseased joint to repair the damage caused by the overload”. I don’t know if I agree with this sentence. Please reword, remove, or provide supporting references. The next sentence about osteophytes is also reductive.

Response: As the reviewer suggested, we have revised this paragraph and removed the sentence. In addition, we have included the references (Ref. 2, 14, 15) that discuss the possible role of these bone remodeling events in OA.

Language/grammar is generally good, but could use some editing in some places.

Response: As the reviewer suggested, we have made the revisions to improve the writing.

Need to define all acronyms at their first use.

Response: As the reviewer suggested, we have added some missing definitions of the acronyms including NSAID, TNF, μ CT, and OARSI.

How/why did the authors choose the cell lines that you did?

Response: We are a lab focused on skeletal biology and we have multiple cell lines available. To examine NGFR expression, we tested these cell lines and some primary cells to understand the expression profiles of NGFR in skeletal cells. Based on the results from the cell lines, we chose C2C12 for mechanistic studies, because C2C12 is widely used in bone biology studies, and it is more technically amenable for cell culture, passages, and virus transductions than many other cell lines.

Please be consistent with using TrkA vs. Ntrk1.

Response: As suggested, we have used TrkA to replace Ntrk1 in the revised manuscript.

All graphs should show individual data points

Response: As suggested, we have added individual data points to the graphs.

Where are the results on OARSI score? This is a primary outcome, but the data is not shown.

Was grading of synovitis performed? It doesn't appear so, but this should definitely be done.

Response: As suggested, we have provided the OARSI score and synovitis score for the NGFR KO mice in the revised manuscript (Supplementary Fig. 7). These data may suggest that NGFR deficiency in skeletal cells has major effects in regulating bone formation in the joint, while having limited effects on synovial inflammation and cartilage degradation at the stage we studied in this work. Thus, the phenotypic changes in cartilage and synovium were not as significant as bone remodeling.

REVIEWERS' COMMENTS

Reviewer #1 (Remarks to the Author):

I have no further comments.

Reviewer #2 (Remarks to the Author):

I thank the authors for their careful and polite attention to my suggestions and have nothing substantial to add. However, I would ask that their new sentence ``Pain associated molecules also modulates the pain-affected non-neuronal tissues' be reduced to ``Pain associated molecules also modulates non-neuronal tissues'. As in my previous comments, the authors do not investigate pain at all in this manuscript, nor do they investigate neurogenic inflammation or any other effects of pain on non-neuronal tissues.

Reviewer #3 (Remarks to the Author):

The authors have been responsive to my previous review, and I have no remaining concerns.

REVIEWER COMMENTS

Reviewer #1 (Remarks to the Author):

I have no further comments.

Reviewer #2 (Remarks to the Author):

I thank the authors for their careful and polite attention to my suggestions and have nothing substantial to add. However, I would ask that their new sentence ``Pain associated molecules also modulates the pain-affected non-neuronal tissues' be reduced to ``Pain associated molecules also modulates non-neuronal tissues'. As in my previous comments, the authors do not investigate pain at all in this manuscript, nor do they investigate neurogenic inflammation or any other effects of pain on non-neuronal tissues.

Reviewer #3 (Remarks to the Author):

The authors have been responsive to my previous review, and I have no remaining concerns.

Response: We thank the reviewers for their positive and constructive comments. As Reviewer # 2 suggested, we have revised the sentence to "Pain associated molecules also modulates non-neuronal tissues". In addition, we have further revised or deleted the sentences such as "how pain signaling regulates non-neural tissues" and "the interaction between pain and skeletal pathophysiology" to avoid this kind of statements as requested by the editor.